# A Large-Scale 3D Face Mesh Video Dataset via Neural Re-parameterized Optimization

**Kim Youwang**                                                      *youwang.kim@postech.ac.kr*
*Department of Electrical Engineering, POSTECH*

**Lee Hyun**$^{*\dagger}$                                            *hyunlee@postech.ac.kr*
*Department of Electrical Engineering, POSTECH*

**Kim Sung-Bin**$^{\dagger}$                                         *sungbin@postech.ac.kr*
*Department of Electrical Engineering, POSTECH*

**Suekyeong Nam**                                                    *sk.nam@krafton.com*
*KRAFTON*

**Janghoon Ju**                                                      *janghoon.ju@krafton.com*
*KRAFTON*

**Tae-Hyun Oh**                                                      *taehyun@postech.ac.kr*
*Department of Electrical Engineering & Grad. School of AI, POSTECH*
*I-CREATE, Yonsei University*

**Reviewed on OpenReview:** *https://openreview.net/forum?id=zVDMh6JvWc*

## Abstract

We propose NeuFace, a 3D face mesh pseudo annotation method on videos via neural re-parameterized optimization. Despite the huge progress in 3D face reconstruction methods, generating reliable 3D face labels for in-the-wild dynamic videos remains challenging. Using NeuFace optimization, we annotate the per-view/-frame accurate and consistent face meshes on large-scale face videos, called the NeuFace-dataset. We investigate how neural re-parameterization helps to reconstruct 3D facial geometries, well complying with input facial gestures and motions. By exploiting the naturalness and diversity of 3D faces in our dataset, we demonstrate the usefulness of our dataset for 3D face-related tasks: improving the reconstruction accuracy of an existing 3D face reconstruction model and learning 3D facial motion prior. Project page: https://kim-youwang.github.io/neuface

## 1 Introduction

A comprehensive understanding of *dynamic* 3D human faces has been a long-standing problem in computer vision and graphics. Reconstructing and generating dynamic 3D human faces are key components for diverse tasks such as face recognition (Weyrauch et al., 2004; Blanz & Vetter, 2003), face forgery detection (Cozzolino et al., 2021; Rössler et al., 2018; 2019), video face editing (B R et al., 2021; Kim et al., 2018; Tewari et al., 2020), facial motion or expression transfer (Thies et al., 2015; 2016a; 2018), XR applications (Elgharib et al., 2020; Wang et al., 2021; Richard et al., 2021; Saito et al., 2024; Sung-Bin et al., 2024a;b; EunGi et al., 2024), and human avatar generation (Raj et al., 2020; Ma et al., 2021; Youwang et al., 2022; 2024).

Recent studies (Wood et al., 2021; 2022; Bae et al., 2023; Yeh et al., 2022) have shown that reliable datasets of facial geometry, even synthetic or pseudo ones, can help achieve a comprehensive understanding of *"static"*

---

$^{*}$ Now at Samsung Advanced Institute of Technology (SAIT).

$^{\dagger}$ Equally contributed 2*nd* authors.

Table 1: **NeuFace-dataset** provides reliable 3D face mesh annotations for MEAD, VoxCeleb2 and CelebV-HQ videos, which is significantly richer than the existing datasets in terms of the scale, diversity and naturalness. *Abbr.* {seq.: sequences, id.: identities, Dur.: duration, Env.: environment, emo.: emotion}

| Dataset | No. seq. [K] | No. id | Dur. [hrs] | Env. | No. emo. |
|---|---|---|---|---|---|
| **Existing 3D face video datasets** | | | | | |
| BIWI 3D | 1.1 | 14 | 1.4 | Lab. | 12 |
| COMA | 0.15 | 12 | 0.1 | Lab. | 12 |
| VOCASET | 0.5 | 12 | 0.5 | Lab. | 1 |
| **NeuFace-dataset (ours)** | **1,245** | **21,048** | **2,090** | **Wild + Lab** | **24** |
| ⊢ NeuFace$_{\text{MEAD}}$ | 210 | 48 | 25 | Lab. | 24 |
| ⊢ NeuFace$_{\text{VoxCeleb2}}$ | 1,000 | 6,000 | 2,000 | Wild | - |
| ⊢ NeuFace$_{\text{CelebV-HQ}}$ | 35 | 15,000 | 65 | Wild | 8 |

3D faces. However, there is currently a lack of reliable and large-scale datasets containing *"dynamic"* and *"natural"* 3D facial motion annotations. The lack of such datasets becomes a bottleneck for studying inherent facial motion dynamics or 3D face reconstruction tasks by restricting them to rely on weak supervision, *e.g.*, 2D landmarks or segmentation maps. Accurately acquired 3D face video data may mitigate such issues but typically requires intensive and time-consuming efforts with carefully calibrated multi-view cameras and controlled lighting conditions (Yoon et al., 2021; Joo et al., 2015; 2018; Cudeiro et al., 2019; Ranjan et al., 2018). Few seminal works (Fanelli et al., 2010; Ranjan et al., 2018; Cudeiro et al., 2019; Zielonka et al., 2022) take such effort to build 3D face video datasets. Despite significant efforts, the existing datasets obtained from such restricted settings are limited in scale, scenarios, diversity of actor identity and expression, and naturalness of facial motion (see Table 1).

In contrast to 3D, there are an incomparably large amount of 2D face video datasets available online (Wang et al., 2020; Nagrani et al., 2017; Chung et al., 2018; Zhu et al., 2022; Parkhi et al., 2015; Cao et al., 2018; Karras et al., 2019; Wang et al., 2021; 2019; Liu et al., 2015), which are captured in diverse in-the-wild environments but without 3D annotations. As successfully demonstrated in some 3D tasks (Fang et al., 2021; Bouazizi et al., 2021; Huang et al., 2022; Müller et al., 2021; Hassan et al., 2019; Bayer et al., 2016; Ng et al., 2022) as well as other analysis tasks (Miech et al., 2019; Nagrani et al., 2022; Lee et al., 2021), leveraging off-the-shelf reconstruction models is a common practice to obtain pseudo ground-truth of such in-the-wild videos that were already captured. They showed that high-quality and large-scale pseudo ground-truth is sufficient to achieve the state-of-the-art at the time of their works. Similarly, a naïve approach is to construct a large-scale 3D face video dataset by curating existing 2D video datasets and obtain 3D face annotations with off-the-shelf face reconstruction models (Feng et al., 2021; Danecek et al., 2022). However, existing 3D face reconstruction models have limitations for reconstructing temporally smooth or multi-view consistent 3D face meshes from videos. This is because state-of-the-art face reconstruction models are typically trained on single-view static images only with 2D supervision; thus fail to extrapolate to faces having rare poses and yield jittered motion due to the per-frame independent inference.

To address these difficulties, we propose **NeuFace optimization**, which reconstructs accurate and spatio-temporally consistent parametric 3D face meshes on videos. By re-parameterizing 3D face meshes with neural network parameters, NeuFace infuses spatio-temporal cues of dynamic face videos on 3D face reconstruction. NeuFace optimizes spatio-temporal consistency losses and the 2D landmark loss to acquire reliable face mesh pseudo-labels for videos.

Using this method, we create the **NeuFace-dataset**, the first large-scale, accurate and spatio-temporally consistent 3D face meshes for videos. Our dataset contains 3D face mesh pseudo-labels for large-scale, multi-view or in-the-wild 2D face videos, MEAD (Wang et al., 2020), VoxCeleb2 (Chung et al., 2018), and CelebV-HQ (Zhu et al., 2022), achieving about 1,000 times larger number of sequences than existing facial motion capture datasets (see Table 1). Our dataset inherits the benefits of the rich visual attributes in large-scale face videos, *e.g.*, various races, appearances, backgrounds, natural facial motions, and expressions. We assess the fidelity of our dataset by investigating the cross-view vertex distance and the 3D motion stability index. We demonstrate that our dataset contains more spatio-temporally consistent and accurate 3D meshes than the competing datasets built with strong baseline methods. To demonstrate the potential of

our dataset, we present two applications: (1) improving the accuracy of a face reconstruction model and (2) learning a generative 3D facial motion prior. These applications highlight that NeuFace-dataset can be further used in diverse applications demanding high-quality and large-scale 3D face meshes. We summarize our main contributions as follows:

- **NeuFace**, an optimization method for reconstructing accurate and spatio-temporally consistent 3D face meshes on videos via neural re-parameterization.
- **NeuFace-dataset**, the first large-scale 3D face mesh pseudo-labels constructed by curating existing large-scale 2D face video datasets with our method.
- Demonstrating the benefits of NeuFace-dataset: (1) improve the accuracy of off-the-shelf face mesh regressors, (2) learn 3D facial motion prior for long-term face motion generation.

## 2 Related work

**3D face datasets.** To achieve a comprehensive understanding of dynamic 3D faces, large-scale in-the-wild 3D face video datasets are essential. There exist large-scale 2D face datasets that provide expressive face images or videos (Wang et al., 2020; Nagrani et al., 2017; Chung et al., 2018; Zhu et al., 2022; Parkhi et al., 2015; Cao et al., 2018; Karras et al., 2019; Liu et al., 2015) with diverse attributes covering a wide variety of appearances, races, environments, scenarios, and emotions. However, most 2D face datasets do not have corresponding 3D annotations, due to the difficulty of 3D face acquisition, especially for in-the-wild environments. Although some recent datasets (Yoon et al., 2021; Ranjan et al., 2018; Cudeiro et al., 2019; Zielonka et al., 2022; Wood et al., 2021) provide 3D face annotations with paired images or videos,[1] they are acquired in the restricted and carefully controlled indoor capturing environment, *e.g.*, laboratory, yielding small scale, unnatural facial expressions and a limited variety of facial identities or features. Achieving in-the-wild naturalness and acquiring true 3D labels would be mutually exclusive in the real-world. Due to the challenge of constructing a real-world 3D face dataset, FaceSynthetics (Wood et al., 2021) synthesizes large-scale synthetic face images and annotations derived from synthetic 3D faces, but limited in that they only publish images and 2D annotations without 3D annotations, which restrict 3D face video applications. In this work, we present the **NeuFace-dataset**, the first large-scale 3D face mesh pseudo-labels paired with the existing in-the-wild 2D face video datasets, resolving the lack of the 3D face video datasets.

**3D face reconstruction.** To obtain reliable face meshes for large-scale face videos, we need accurate 3D face reconstruction methods for videos. Reconstructing accurate 3D faces from limited visual cues, *e.g.*, a monocular image, is an ill-posed problem. Model-based approaches have been the mainstream to mitigate the ill-posedness and have advanced with the 3D Morphable Models (3DMMs) (Blanz & Vetter, 1999; Paysan et al., 2009; Li et al., 2017) and 3DMM-based reconstruction methods (Zollhöfer et al., 2018; Egger et al., 2020; Feng et al., 2021; Danecek et al., 2022; Zielonka et al., 2022).

3D face reconstruction methods can be categorized into learning-based and optimization-based approaches. The learning-based approaches, *e.g.*, (Feng et al., 2021; Danecek et al., 2022; Zielonka et al., 2022; Sanyal et al., 2019a; an Trần et al., 2016), use neural networks trained on large-scale face image datasets to regress the 3DMM parameters from a single image. The optimization-based approaches (Blanz & Vetter, 2003; Huber et al., 2015; Chen et al., 2013; Wood et al., 2022; Thies et al., 2015; Gecer et al., 2019) optimize the 2D landmark or photometric losses with extra regularization terms directly over the 3DMM parameters. Given a specific image, these methods overfit to 2D landmarks observations, thus showing better 2D landmark fit than the learning-based methods. These approaches are suitable for our purpose in that we need accurate reconstruction that best fits each video. However, the regularization terms are typically hand-designed with prior assumptions that disregard the input image. These regularization terms often introduce mean shape biases (Feng et al., 2021; Pavlakos et al., 2019; Bogo et al., 2016; Joo et al., 2020), due to their independence to input data, which we call the *data-independent prior*. Also, balancing the losses and regularization is inherently cumbersome and may introduce initialization sensitivity and local minima issues (Joo et al., 2020; Pavlakos et al., 2019; Bogo et al., 2016; Choutas et al., 2020).

---

[1]MICA released the medium-scale 3D annotated face datasets, but only a single identity parameter per video is provided, not the facial poses or expression parameters, *i.e.*, static 3D faces.

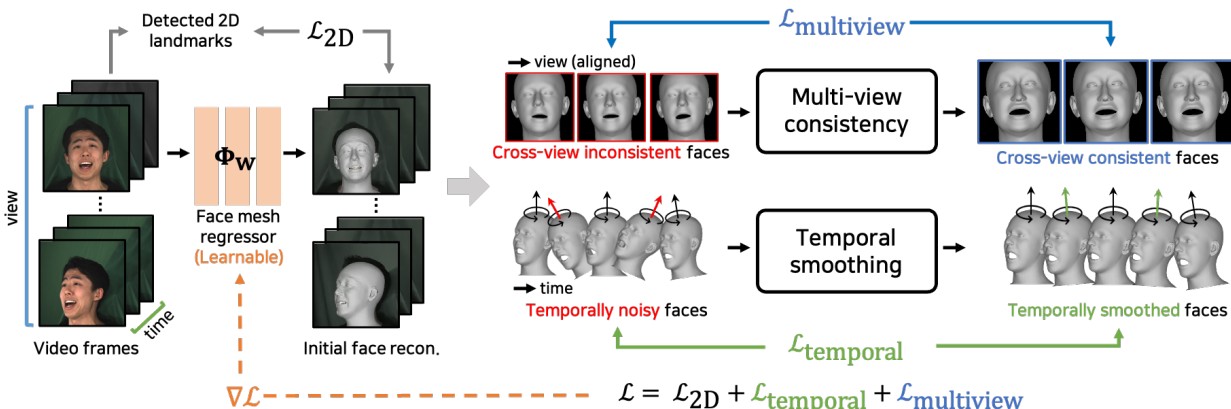

Figure 1: **NeuFace optimization.** Given 2D face videos, NeuFace optimizes spatio-temporally consistent 3D face meshes. NeuFace updates the neural network parameters that re-parameterize the 3D face meshes with 2D landmark loss and spatio-temporal consistency losses.

Instead of hand-designed regularization terms, we induce such effects by optimizing re-parameterized 3DMM parameters with a 3DMM regression neural network, called **NeuFace** optimization. Such network parameters are trained from large-scale real face images, which implicitly embed strong prior from the trained data. Thereby, we can leverage the favorable properties of the neural re-parameterization: 1) an input *data-dependent* initialization and prior in 3DMM parameter optimization, 2) less bias toward a mean shape, and 3) stable optimization robust to local minima by over-parameterized model (Cooper, 2021; Du et al., 2019a; Neyshabur et al., 2018; Allen-Zhu et al., 2019; Du et al., 2019b; Youwang et al., 2024). Similar re-parameterizations were proposed in (Joo et al., 2020; Grassal et al., 2022), but they focus on the human body in a single image input with fixed 2D landmark supervision, or use MLP to re-parameterize the per-vertex displacement of the 3D face. We extend it to dynamic faces in the multi-view and video settings by sharing the neural parameters across views and frames, and devise an alternating optimization to self-supervise spatio-temporal consistency.

## 3 NeuFace: a 3D face mesh optimization for videos via neural re-parameterization

In this section, we introduce the neural re-parameterization of 3DMM (Sec. 3.1) and NeuFace, an optimization to obtain accurate and spatio-temporally consistent face meshes from videos (Sec. 3.2). We discuss the benefit of neural re-parameterization (Sec. 3.3), and show NeuFace's reliability as a face mesh annotator (Sec. 3.4).

### 3.1 Neural re-parameterization of 3D face meshes

We use FLAME (Li et al., 2017), a renowned 3DMM, as a 3D face representation. 3D face mesh vertices $\mathbf{M}$ and facial landmarks $\mathbf{J}$ for $F$ frame videos can be acquired with the differentiable skinning: $\mathbf{M}, \mathbf{J} = \text{FLAME}(\mathbf{r}, \boldsymbol{\theta}, \boldsymbol{\beta}, \boldsymbol{\psi})$, where $\mathbf{r} \in \mathbb{R}^3$, $\boldsymbol{\theta} \in \mathbb{R}^{12}$, $\boldsymbol{\beta} \in \mathbb{R}^{100}$ and $\boldsymbol{\psi} \in \mathbb{R}^{50}$ denote the head orientation, face poses, face shape and expression coefficients, respectively. For simplicity, FLAME parameters $\boldsymbol{\Theta}$ can be represented as, $\boldsymbol{\Theta} = [\mathbf{r}, \boldsymbol{\theta}, \boldsymbol{\beta}, \boldsymbol{\psi}]$. We further re-parameterize the FLAME parameters $\boldsymbol{\Theta}$ and weak perspective camera parameters $\mathbf{p} \in \mathbb{R}^{F \times 3}$ for video frames $\{\mathbf{I}_f\}_{f=1}^F$, into a neural network, $\Phi$, with parameters $\mathbf{w}$, *i.e.*, $[\boldsymbol{\Theta}, \mathbf{p}] = \Phi_{\mathbf{w}}(\{\mathbf{I}_f\}_{f=1}^F)$. We use the pre-trained DECA (Feng et al., 2021) or EMOCA (Danecek et al., 2022) encoder for $\Phi_{\mathbf{w}}$.

### 3.2 NeuFace optimization

Given the $N_F$ frames and $N_V$ views of a face video $\{\mathbf{I}_{f,v}\}_{f=1,v=1}^{N_F,N_V}$, NeuFace aims to find the optimal neural network parameter $\mathbf{w}^*$ that re-parameterizes accurate, multi-view and temporally consistent face meshes (see Fig. 1). The optimization objective is defined as:

$$\mathbf{w}^* = \arg\min_{\mathbf{w}} \ \mathcal{L}_{\text{2D}} + \lambda_{\text{temp}}\mathcal{L}_{\text{temporal}} + \lambda_{\text{view}}\mathcal{L}_{\text{multiview}}, \tag{1}$$

where $\{\lambda_*\}$ denotes the weights for each loss term. Complex temporal and multi-view dependencies among variables in the losses would make direct optimization difficult (Afonso et al., 2010; Salzmann, 2013; Zhang, 1993). We ease the optimization of Eq. (1) by introducing latent target variables for self-supervision in an Expectation-Maximization (EM) style optimization (see Sec. B for the details of EM style optimization).

**2D landmark loss.** For each iteration $t$, we compute $\mathcal{L}_{2D}$ as a unary term, following the conventional 2D facial landmark re-projection loss (Feng et al., 2021; Danecek et al., 2022) for the landmarks in all different frames and views:

$$\mathcal{L}_{2D} = \frac{1}{N_F N_V} \sum_{f=1,v=1}^{N_F, N_V} \|\pi(\mathbf{J}_{f,v}^t(\mathbf{w}), \mathbf{p}_{f,v}^t) - \mathbf{j}_{f,v}\|_1, \tag{2}$$

where $\pi(\cdot, \cdot)$ denotes the weak perspective projection, and $\mathbf{J}(\mathbf{w})$ is the 3D landmark from $\Phi_{\mathbf{w}}(\cdot)$. Eq. (2) computes the pixel distance between the pre-detected 2D facial landmarks $\mathbf{j}$ and the regressed and projected 3D facial landmarks $\pi(\mathbf{J}(\mathbf{w}), \mathbf{p})$. $\mathbf{j}$ stays the same for the whole optimization. We use FAN (Bulat & Tzimiropoulos, 2017) to obtain $\mathbf{j}$ with human verification to reject the failure cases.

**Temporal consistency loss.** Our temporal consistency loss reduces facial motion jitter caused by per-frame independent mesh regression on videos. Instead of a complicated Markov chain style loss, for each iteration $t$, we first estimate latent target meshes that represent temporally smooth heads in Expectation step (E-step). Then, we simply maximize the likelihood of regressed meshes to its corresponding latent target in Maximization step (M-step). In E-step, we feed $\{\mathbf{I}_{f,v}\}_{f=1,v=1}^{N_F, N_V}$ into the network $\Phi_{\mathbf{w}^t}$ and obtain FLAME and camera parameters, $[\Theta^t, \mathbf{p}^t]$. For multiple frames in view $v$, we extract the head orientations $\mathbf{r}_{:,v}^t$, from $\Theta^t$ and convert it to the unit quaternion $\mathbf{q}_{:,v}^t$. To generate the latent target, *i.e.*, temporally smooth head orientations $\hat{\mathbf{q}}_{:,v}^t$, we take the temporal moving average over $\mathbf{q}_{:,v}^t$. In M-step, we compute the temporal consistency loss as:

$$\mathcal{L}_{temporal} = \frac{1}{N_F N_V} \sum_{f=1,v=1}^{N_F, N_V} \|\mathbf{q}_{f,v}^t - \hat{\mathbf{q}}_{f,v}^t\|_2, \tag{3}$$

where $\mathbf{q}$ is the unit-quaternion representation of $\mathbf{r}$. We empirically found that such simple consistency loss is sufficient enough to obtain temporal smoothness while allowing more flexible expressions.

**Multi-view consistency loss.** Although the aforementioned $\mathcal{L}_{2D}$ roughly guides the multi-view consistency of landmarks, it cannot guarantee the consistency for off-landmark or invisible facial regions across views. Therefore, for multi-view captured face videos (Wang et al., 2020), we leverage a simple principle to obtain consistent meshes over different views: face geometry should be consistent across views at the same time. The goal is to bootstrap the per-view estimated noisy meshes by referencing the visible, or highly confident facial regions across different views. Analogous to the temporal consistency loss, in M-step, we compute the multi-view consistency loss as follows:

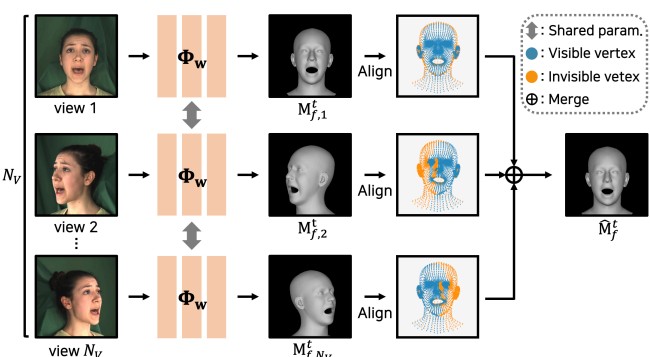

Figure 2: **Multi-view bootstrapping.** Given initial mesh predictions for each view in frame $f$, we align and merge the meshes depending on the confidence. The bootstrapped mesh serves as a target for computing $\mathcal{L}_{multiview}$.

$$\mathcal{L}_{multiview} = \frac{1}{N_F N_V} \sum_{f=1,v=1}^{N_F, N_V} \|\mathbf{M}_{f,v}^t - \hat{\mathbf{M}}_f^t\|_1, \tag{4}$$

where $\hat{\mathbf{M}}_f^t$ denotes the latent target mesh vertices estimated in E-step of each iteration. In E-step, given vertices $\mathbf{M}_{f,:}^t$ of multiple views in frame $f$, we interpret the vertex visibility as the per-vertex confidence. We assign the confidence score per each vertex by measuring the angle between the vertex normal and the camera ray. We set the vertices as invisible if the angle is larger than the threshold $\tau_a$, and the vertex has a deeper depth than $\tau_z$, *i.e.*, $z < \tau_z$. We empirically choose $\tau_a = 72°$, $\tau_z = -0.08$. To obtain the latent target mesh $\hat{\mathbf{M}}_f^t$, we align per-view estimated meshes to the canonical view, and bootstrap the meshes by taking the weighted average of $\mathbf{M}_{f,:}^t$ depending on the confidence (see Fig. 2). With this, Eq. (4) constrains the vertices of each view to be consistent with $\hat{\mathbf{M}}_f^t$.

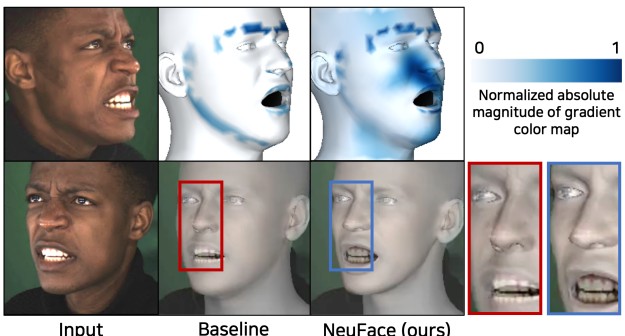

Figure 3: **Data-dependent gradient.** NeuFace optimization obtains a richer gradient map regarding the pixel-level facial details (1*st* row). Thus, our method achieves more expressive and accurately meshes than the baseline (2*nd* row).

**Overall process.** We first estimate all the latent variables, $\hat{\mathbf{q}}$ and $\hat{\mathbf{M}}$ as E-step. With the estimated latent variables as the self-supervision target, we optimize Eq. (1) over the *network parameter* $\mathbf{w}$ as M-step. This single alternating iteration updates the optimization parameter $\mathbf{w}^t \rightarrow \mathbf{w}^{t+1}$ at iteration $t$. We iterate alternating E-step and M-step until convergence. After convergence, we obtain the final solution $[\boldsymbol{\Theta}^*, \mathbf{p}^*]$ by querying video frames to the optimized network, *i.e.*, $[\boldsymbol{\Theta}^*, \mathbf{p}^*] = \Phi_{\mathbf{w}^*}(\{\mathbf{I}_{f,v}\}_{f=1,v=1}^{N_F,N_V})$.

### 3.3 Why is NeuFace optimization effective?

Note that one can simply update FLAME parameters directly with the same loss in Eq. (1). Then, why do we need neural re-parameterization of 3D face meshes? We claim such neural re-parameterization allows *data-dependent mesh update*, which the FLAME fitting cannot achieve. To support our claim, we analyze the benefit of our optimization by comparing it with the solid baseline.

**Baseline: FLAME fitting.** Given the same video frames $\{\mathbf{I}_{f,v}\}_{f=1,v=1}^{N_F,N_V}$ and the same initial FLAME and camera parameters $[\boldsymbol{\Theta}_{\mathbf{b}}, \mathbf{p}_{\mathbf{b}}]$ as NeuFace[2], we implement the baseline optimization as:

$$[\boldsymbol{\Theta}_{\mathbf{b}}^*, \mathbf{p}_{\mathbf{b}}^*] = \underset{\boldsymbol{\Theta}_{\mathbf{b}}, \mathbf{p}_{\mathbf{b}}}{\arg\min} \ \mathcal{L}_{\text{2D}} + \lambda_{\text{temp}}\mathcal{L}_{\text{temporal}} + \lambda_{\text{view}}\mathcal{L}_{\text{multiview}} + \lambda_{\mathbf{r}}\mathcal{L}_{\mathbf{r}} + \lambda_{\boldsymbol{\theta}}\mathcal{L}_{\boldsymbol{\theta}} + \lambda_{\boldsymbol{\beta}}\mathcal{L}_{\boldsymbol{\beta}} + \lambda_{\boldsymbol{\psi}}\mathcal{L}_{\boldsymbol{\psi}}, \tag{5}$$

where the losses $\mathcal{L}_{\text{2D}}$, $\mathcal{L}_{\text{temporal}}$ and $\mathcal{L}_{\text{multiview}}$ are identical to the Eqs. (2), (3), and (4). $\mathcal{L}_{\mathbf{r}}$, $\mathcal{L}_{\boldsymbol{\theta}}$, $\mathcal{L}_{\boldsymbol{\beta}}$ and $\mathcal{L}_{\boldsymbol{\psi}}$, are the common regularization terms used in (Li et al., 2017; Wood et al., 2022).

**Data-dependent gradients for mesh update.** We analyze the data-dependency of the baseline and NeuFace optimization by investigating back-propagated gradients. For the FLAME fitting (Eq. (5)), the update rule for FLAME parameters $\boldsymbol{\Theta}_{\mathbf{b}}$ at optimization step $t$ is as follows:

$$\boldsymbol{\Theta}_{\mathbf{b}}^{t+1} = \boldsymbol{\Theta}_{\mathbf{b}}^t - \alpha\frac{\partial\mathcal{L}}{\partial\boldsymbol{\Theta}_{\mathbf{b}}^t}, \tag{6}$$

where $\mathcal{L}$ denotes the sum of all the losses used in the optimization. In contrast, given video frames $\{\mathbf{I}_{f,v}\}_{f=1,v=1}^{N_F,N_V}$, or simply $\mathbf{I}$, the update for our NeuFace optimization is as follows:

$$\mathbf{w}^{t+1} = \mathbf{w}^t - \alpha\frac{\partial\mathcal{L}}{\partial\mathbf{w}^t} = \mathbf{w}^t - \alpha\big(\frac{\partial\mathcal{L}}{\partial\boldsymbol{\Theta}_{\mathbf{w}}^t} \cdot \frac{\partial\boldsymbol{\Theta}_{\mathbf{w}}^t}{\partial\mathbf{w}^t}\big) = \mathbf{w}^t - \alpha\big(\frac{\partial\mathcal{L}}{\partial\boldsymbol{\Theta}_{\mathbf{w}}^t} \cdot \frac{\partial}{\partial\mathbf{w}^t}\Phi_{\mathbf{w}^t}(\mathbf{I})\big), \tag{7}$$

where $\boldsymbol{\Theta}_{\mathbf{w}}^t$ is re-/over-parameterized by the neural network $\Phi_{\mathbf{w}^t}$, *i.e.*, $\boldsymbol{\Theta}_{\mathbf{w}}^t = \Phi_{\mathbf{w}^t}(\mathbf{I})$.

By comparing the back-propagated gradient terms in Eqs. (6) and (7), we can intuitively notice that the update for NeuFace optimization (Eq. (7)) is conditioned by input $\mathbf{I}$, yielding *data-dependent* mesh update. With data-dependent gradient $\frac{\partial}{\partial\mathbf{w}^t}\Phi_{\mathbf{w}^t}(\mathbf{I})$, NeuFace optimization may inherit the implicit prior embedded in the pre-trained neural model, *e.g.*, DECA (Feng et al., 2021), learned from large-scale real face images.

---

[2]To conduct a fair comparison with a strong baseline, we initialize $[\boldsymbol{\Theta}_{\mathbf{b}}, \mathbf{p}_{\mathbf{b}}]$ as the prediction of the pre-trained DECA. This is identical to NeuFace optimization (Eq. (1)); only the optimization variable is different.

This allows NeuFace optimization to obtain expressive 3D facial geometries, well complying with input facial gestures and motions.

It is also worthwhile to note that, thanks to over-parameterization of $\Phi_{\mathbf{w}}(\cdot)$ *w.r.t.* $\mathbf{\Theta}$, we benefit from the following favorable property. For simplicity, we consider a simple $l_2$-loss and a fully connected ReLU network,[3] but it is sufficient to understand the mechanism of NeuFace optimization.

**Proposition 1** (Informal). **Global convergence.** *For the input data* $\mathbf{x} \in [0,1]^{n \times d_{in}}$*, paired labels* $\mathbf{y}^* \in \mathbb{R}^{n \times d_{out}}$*, and an over-parameterized L-layer fully connected network* $\Phi_{\mathbf{w}}(\cdot)$ *with ReLU activation and uniform weight widths, consider optimizing the non-convex problem:* $\arg\min_{\mathbf{w}} \mathcal{L}(\mathbf{w}) = \frac{1}{2} \|\Phi_{\mathbf{w}}(\mathbf{x}) - \mathbf{y}^*\|_2^2$*. Under some assumptions, gradient descent finds a global optimum in polynomial time with high probability.*

Proposition 1 can be derived by simply re-compositing the results by Allen-Zhu et al. (2019). Its proof sketch can be found in the appendix. This hints that our over-parameterization helps NeuFace optimization achieve robustness to local minima and avoid mean shape biases.

To see how data-dependent gradient of NeuFace affects the mesh optimization, we visualize the absolute magnitude of the back-propagated gradients of each method in Fig. 3. The baseline optimization produces a sparse gradient map along the face landmarks, which disregards the pixel-level facial details, *e.g.*, wrinkles or facial boundaries. In contrast, NeuFace additionally induces the dense gradients over face surfaces, not just sparse landmarks, which are helpful for representing image-aligned and detailed facial expressions on meshes. Thanks to the rich gradient map, our method yields more expressive and accurately image-aligned meshes than the baseline.

### 3.4 How reliable is NeuFace optimization?

Many recent face-related applications (Ng et al., 2022; Khakhulin et al., 2022; Feng et al., 2022) utilize a pre-trained, off-the-shelf 3D face reconstruction model or the FLAME fitting (Eq. (5)) as a pseudo ground-truth annotator. Compared to such conventional face mesh annotation methods, we discuss how reliable Neuface optimization is. Specifically, we measure the vertex-level accuracy of the reconstructed face meshes by NeuFace optimization on the motion capture videos, VOCASET (Cudeiro et al., 2019).

VOCASET is a small-scale facial motion capture dataset that provides registered ground-truth mesh sequences. Given the ground-truth mesh sequences from the VOCASET, we evaluate the Mean-Per-Vertex-Error (MPVE) (Cho et al., 2022; Lin et al., 2021b;a) of face meshes obtained by pre-trained DECA, FLAME fitting and our method. In Fig. 4, NeuFace optimization achieves more vertex-level accurate meshes than other methods, *i.e.*, lower MPVE. Note that FLAME fitting still achieves competitive MPVE with ours, which shows that it is a valid, strong baseline. Such favorable mesh accuracy of NeuFace optimization motivates us to leverage it as a reliable face mesh annotator for large-scale face videos, and build the NeuFace-dataset.

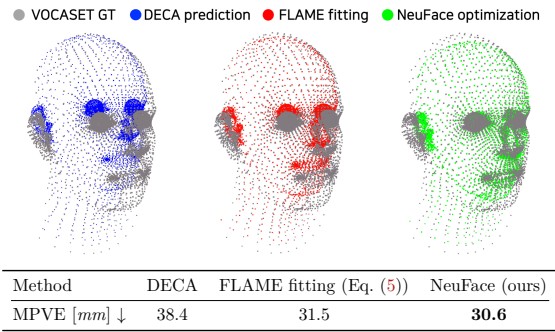

● VOCASET GT ● DECA prediction ● FLAME fitting ● NeuFace optimization

| Method | DECA | FLAME fitting (Eq. (5)) | NeuFace (ours) |
|---|---|---|---|
| MPVE [$mm$] ↓ | 38.4 | 31.5 | **30.6** |

Figure 4: Given the ground-truth meshes, our optimization reconstructs more vertex-level accurate meshes than the competing methods.

## 4 The NeuFace-dataset

The NeuFace-dataset provides accurate and spatio-temporally consistent face meshes of existing large-scale 2D face video datasets; MEAD (Wang et al., 2020), VoxCeleb2 (Chung et al., 2018), and CelebV-HQ (Zhu et al., 2022). Our datasets are denoted with NeuFace$_{\{*\}}$ and summarized in Table 1. The NeuFace-dataset is, namely, the largest 3D face mesh pseudo-labeled dataset in terms of the scale, naturalness, and diversity of facial attributes, emotions, and backgrounds. Please refer to the appendix for the visualization of the dataset, acquisition method, and filtering details.

---

[3]By sacrificing the complexity of proof, the same conclusion holds for ResNet (Allen-Zhu et al., 2019).

Table 2: **Quantitative evaluation.** NeuFace-D/E-datasets (ours) significantly outperform the other datasets in multi-view consistency (CVD), temporal consistency ($MSI_{3D}$), and the 2D landmark accuracy (NME). *Abbr.* {L: landmark, V: vertex.}

| Dataset | MEAD | | | | VoxCeleb2 | | | CelebV-HQ | | |
|---|---|---|---|---|---|---|---|---|---|---|
| | $MSI_{3D}^L \uparrow$ | $MSI_{3D}^V \uparrow$ | CVD $\downarrow$ | NME $\downarrow$ | $MSI_{3D}^L \uparrow$ | $MSI_{3D}^V \uparrow$ | NME $\downarrow$ | $MSI_{3D}^L \uparrow$ | $MSI_{3D}^V \uparrow$ | NME $\downarrow$ |
| Base-dataset (Eq. (5)) | 0.034 | 0.053 | 0.192 | 4.34 | 0.034 | 0.056 | 3.32 | 0.030 | 0.047 | 3.65 |
| DECA-dataset | 0.011 | 0.016 | 0.209 | 4.65 | 0.028 | 0.044 | 4.78 | 0.012 | 0.018 | 5.34 |
| **NeuFace-D-dataset** (ours) | **0.206** | **0.305** | **0.103** | **2.58** | **0.095** | **0.137** | **2.19** | **0.054** | **0.074** | **2.55** |
| EMOCA-dataset | 0.010 | 0.016 | 0.199 | 5.42 | 0.003 | 0.004 | 4.77 | 0.005 | 0.007 | 5.57 |
| **NeuFace-E-dataset** (ours) | **0.209** | **0.312** | **0.104** | **2.28** | **0.028** | **0.048** | **2.38** | **0.053** | **0.077** | **2.86** |

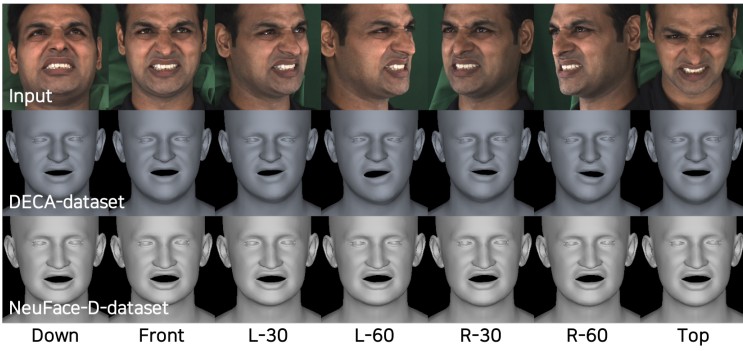

Figure 5: **Multi-view consistent face meshes.** NeuFace-dataset contains multi-view consistent meshes compared to the DECA-dataset. L- and R- denote Left and Right, and 30 and 60 denote the camera view angles from the center.

## 4.1 Fidelity evaluation of the NeuFace-dataset

We assess the fidelity of our dataset in terms of spatio-temporal consistency and landmark accuracy. We make competing datasets and compare the quality of the generated mesh annotations. First, we compose the strong baseline, Base-dataset, by fitting FLAME with Eq. (5). We also utilize pre-trained DECA and EMOCA as mesh annotators and built DECA-dataset and EMOCA-dataset, respectively. Finally, we build two versions of our dataset, *i.e.*, NeuFace-D, and NeuFace-E, where each dataset is generated via Eq. (1) with DECA and EMOCA for the neural re-parameterization $\Phi_{\mathbf{w}}$, respectively.

**Temporal consistency.** We extend the Motion Stability Index (MSI) (Ling et al., 2022) to $MSI_{3D}$ and evaluate the temporal consistency of each dataset. $MSI_{3D}$ computes a reciprocal of the motion acceleration variance of either 3D landmarks or vertices and quantifies facial motion stability for a given $N_F$ frame video, $\{\mathbf{I}_f\}_{f=1}^{N_F}$, as $MSI_{3D}(\{\mathbf{I}_f\}_{f=1}^{N_F}) = \frac{1}{K} \sum_i \frac{1}{\sigma(\mathbf{a}^i)}$, where $\mathbf{a}^i$ denotes the 3D motion acceleration of $i$-th 3D landmarks or vertices, $\sigma(\cdot)$ the temporal variance, and $K$ the number of landmarks or vertices. If the mesh sequence has small temporal jittering, *i.e.*, low motion variance, it has a high $MSI_{3D}$ value. We compute $MSI_{3D}$ for landmarks and vertices, *i.e.*, $MSI_{3D}^L$ and $MSI_{3D}^V$, respectively. Table 2 shows the $MSI_{3D}^L$ and $MSI_{3D}^V$ averaged over the validation sets. For the VoxCeleb2 and CelebV-HQ splits, the NeuFace-D/E-dataset outperform the other datasets in both $MSI_{3D}$s. Remarkably, we have improvements on $MSI_{3D}$ more than 20 times in MEAD. We postulate that the multi-view consistency loss also strengthens the temporal consistency for MEAD. In other words, our losses would be mutually helpful when jointly optimized. We discuss it through loss ablation studies in the later section (Sec. 4.2).

**Multi-view consistency.** We visualize the predicted meshes over different views in Fig. 5, where per-view independent estimations are presented, not a single merged one. We verify that the NeuFace-D-dataset contains multi-view consistent meshes compared to the DECA-dataset, especially near the mouth region. See appendix for the comparison of the EMOCA-dataset and NeuFace-E-dataset. As a quantitative measure, we compute the cross-view vertex distance (CVD), *i.e.*, the vertex distance between two different views, $i$ and $j$, in the same frame $f$: $\|\mathbf{M}_{f,i} - \mathbf{M}_{f,j}\|_1$. We compare the averaged CVD of all views in Table 2. CVD is

Table 3: **Ablation results on the different loss functions.** We evaluate the effect of our proposed spatio-temporally consistent losses by changing the configurations of the loss combinations. Optimizing full loss functions shows favorable results on CVD and NME while outperforming MSI compared to other configurations. We cannot analyze the effect of $\mathcal{L}_{\text{multiview}}$ for VoxCeleb2 and CelebV-HQ since they are taken in the single-camera setup. *Abbr.* L: landmark, V: vertex.

| NeuFace | | | MEAD (Wang et al., 2020) | | | | VoxCeleb2 (Chung et al., 2018) | | | CelebV-HQ (Zhu et al., 2022) | | |
|---|---|---|---|---|---|---|---|---|---|---|---|---|
| $\mathcal{L}_{\text{2D}}$ | $\mathcal{L}_{\text{multiview}}$ | $\mathcal{L}_{\text{temporal}}$ | CVD ↓ | $\text{MSI}_{\text{3D}}^{\text{L}}$ ↑ | $\text{MSI}_{\text{3D}}^{\text{V}}$ ↑ | NME ↓ | $\text{MSI}_{\text{3D}}^{\text{L}}$ ↑ | $\text{MSI}_{\text{3D}}^{\text{V}}$ ↑ | NME ↓ | $\text{MSI}_{\text{3D}}^{\text{L}}$ ↑ | $\text{MSI}_{\text{3D}}^{\text{V}}$ ↑ | NME ↓ |
| ✓ | | | 0.333 | 0.015 | 0.022 | **2.44** | 0.001 | 0.001 | 2.71 | 0.001 | 0.003 | 3.88 |
| ✓ | ✓ | | 0.110 | 0.015 | 0.023 | 2.49 | - | - | - | - | - | - |
| ✓ | | ✓ | 0.294 | 0.176 | 0.250 | 2.56 | **0.095** | **0.137** | **2.19** | **0.54** | **0.074** | **2.55** |
| ✓ | ✓ | ✓ | **0.103** | **0.206** | **0.305** | 2.58 | - | - | - | - | - | - |
| DECA (Feng et al., 2021) | | | 0.209 | 0.011 | 0.016 | 4.65 | 0.028 | 0.044 | 4.78 | 0.012 | 0.018 | 5.34 |

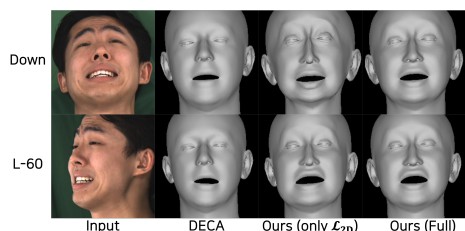

Figure 6: **Effect of the $\mathcal{L}_{\text{multiview}}$.** By optimizing our full objectives (Eq. (1)), the generated meshes are fitted to each of its 2D landmarks and are also multi-view consistent. Views are aligned for the visualization.

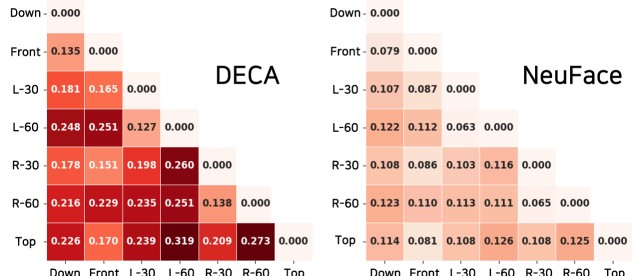

Figure 7: **Comparisons of cross-view vertex distance.** We quantitatively show the multi-view consistency of our method by averaging the cross-view vertex distance on the validation set. L- and R- denote Left and Right, respectively, and 30 and 60 denote the view angles which the video is captured from.

only evaluated on the MEAD dataset, which is in a multi-camera setup. While the DECA-/EMOCA-dataset results in high CVD, the NeuFace-dataset shows significantly lower CVD on overall views.

**2D landmark accuracy.** A trivial solution to obtain low CVD and high $\text{MSI}_{\text{3D}}$ is to regress the same mean face meshes across views and frames regardless of the input image. To verify such occurrence, we measure the landmark accuracy of the regressed 2D facial landmarks using the normalized mean error (NME) (Sagonas et al., 2016). The NeuFace-D/E-dataset outperform the other datasets in NME, *i.e.*, contain spatio-temporally consistent and accurately landmark-aligned meshes.

## 4.2 Ablation on loss functions

We analyze the effect of our proposed spatio-temporal consistency losses in the NeuFace optimization, $\mathcal{L}_{\text{multiview}}$, and $\mathcal{L}_{\text{temporal}}$. We evaluate the quality of the meshes obtained by optimizing each of the loss configurations. All the experiments are conducted on the same validation set as the Table 2.

**Only $\mathcal{L}_{\text{2D}}$.** We start by optimizing the loss function with only 2D facial landmark re-projection error, $\mathcal{L}_{\text{2D}}$. As we optimize $\mathcal{L}_{\text{2D}}$, the obtained meshes achieve lower NME than DECA over the whole validation set (see Table 3). However, we observe that this may break both multi-view and temporal consistencies, degrading the CVD and MSI compared to that of DECA. The qualitative results in Fig. 6 also show that optimizing only $\mathcal{L}_{\text{2D}}$ reconstructs more expressive but inconsistent meshes over different views. Therefore, we propose loss functions that can induce multi-view and temporal consistencies to the meshes during the optimization.

**$\mathcal{L}_{\text{2D}}+\mathcal{L}_{\text{multiview}}$.** By optimizing $\mathcal{L}_{\text{multiview}}$ along with $\mathcal{L}_{\text{2D}}$, we obtain significantly lower CVD than DECA and the meshes optimized with only $\mathcal{L}_{2D}$. We have not optimized any regularization term which induces temporal consistency; thus, MSIs remain low. As discussed in Sec. 4.1, a trivial solution for achieving low

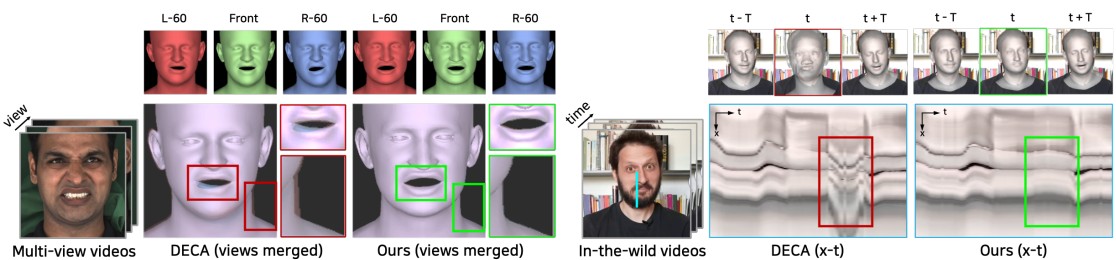

Figure 8: **Spatio-temporal consistency comparison.** NeuFace reconstructs *multi-view consistent 3D faces* (left); reduced misalignment for faces (each view is aligned to the canonical pose, color-coded and alpha-blended), and *temporally stabilized motion* (right); reduced jitter than the competing method (we concatenate vertical cyan line in each frame along time), while being accurate.

CVD is to regress mean faces over different views. However, the optimization with $\mathcal{L}_{2D}+\mathcal{L}_{multiview}$ achieves low CVD while comparable NME to the $\mathcal{L}_{2D}$ optimization, which proves not to be falling into a trivial solution. Note that $\mathcal{L}_{multiview}$ can only be measured in the multi-camera setup, *e.g.* MEAD.

**$\mathcal{L}_{2D}+\mathcal{L}_{temporal}$.** As we optimize $\mathcal{L}_{temporal}$ with $\mathcal{L}_{2D}$, we observe substantial improvements in MSIs over the whole validation set. Interestingly, jointly optimizing these two losses can further achieve better NME in VoxCeleb2 and CelebV-HQ datasets. We postulate that, for in-the-wild challenging cases, *e.g.*, images containing extreme head poses or diverse background scenes, only optimizing $\mathcal{L}_{2D}$ could fail to regress proper meshes, as it may generate meshes that break out of the facial regions. On the other hand, $\mathcal{L}_{temporal}$ could prevent the regressed mesh from breaking out from the facial regions to a certain extent.

**Full loss function.** With the observations of the effect on each proposed loss, we optimize the full loss functions (NeuFace optimization), $\mathcal{L}_{2D}+\mathcal{L}_{spatial}+\mathcal{L}_{temporal}$, on MEAD. The quantitative results in Table 3 show that NeuFace optimization achieves comparable NME while outperforming CVD and MSIs compared to other settings. As analyzed in Sec. 4.1, we postulate that our proposed spatio-temporal losses are mutually helpful for generating multi-view and temporally consistent meshes.

The advantage of jointly optimizing all the losses can also be found in the qualitative results; the reconstructed face meshes are well-fitted to its 2D face features, *e.g.*, landmarks and wrinkles, and multi-view consistent (See Fig. 6). In addition, we compare the view-wise averaged CVD between NeuFace optimization and DECA in Fig. 7. While DECA results in high CVD, especially for the views with self-occluded regions, such as Left-60 and Right-60, NeuFace shows significantly lower CVD on overall views. We also evaluate the meshes reconstructed by pre-trained DECA (Feng et al., 2021) for comparison (see Fig. 8).

## 5 Applications of the NeuFace-datasets

In this section, we demonstrate the usefulness of the NeuFace-dataset. We learn generative facial motion prior from the large-scale, in-the-wild 3D faces in our dataset (Sec. 5.1). Also, we boost the accuracy of an off-the-shelf face mesh regressor by exploiting our dataset's 3D supervision (Sec. 5.2).

### 5.1 Learning 3D human facial motion prior

A facial motion prior is a versatile tool to understand how human faces move over time. It can generate realistic motions or regularize temporal 3D reconstruction (Rempe et al., 2021). Unfortunately, the lack of large-scale 3D face video datasets makes learning facial motion prior infeasible. We tackle this by exploiting the scale, diversity, and naturalness of the 3D facial motions in our dataset.

**Learning facial motion prior.** We learn a 3D facial motion prior using HuMoR (Rempe et al., 2021) with simple modifications. HuMoR is a conditional VAE (Sohn et al., 2015) that learns the transition distribution of human body motion. We represent the state of a facial motion sequence as the combination

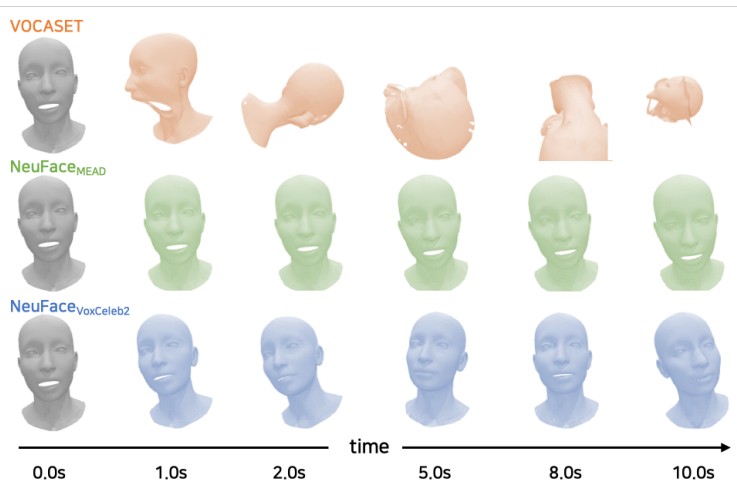

Figure 9: **Long-term facial motion generation using learned motion prior.** The motion prior trained with small-scale, diversity-limited VOCASET fails to generate natural motion, while the motion prior trained with NeuFace$_{\mathrm{VoxCeleb2}}$ generates diverse and natural long-term facial motion.

Table 4: **Quantitative evaluation of learned facial motion prior.** We evaluate the *naturalness* and *diversity* of generated long-term motions from different motion prior models. HuMoR-Face trained with existing facial motion capture dataset, *e.g.*, VOCASET, fails to generate natural and diverse facial motions.

| HuMoR-Face trained w/ | Scale | Environment | FD ↓ | APD [*cm*] ↑ |
|---|---|---|---|---|
| **Existing motion capture dataset** | | | | |
| VOCASET (Cudeiro et al., 2019) | Small | In-the-lab | 420.92 | - |
| **Our dataset** | | | | |
| NeuFace$_{\mathrm{MEAD}}$ | Large | In-the-lab | 78.99 | 3.56 |
| NeuFace$_{\mathrm{VoxCeleb2}}$ | Large | **In-the-wild** | **31.32** | **52.69** |

of FLAME parameters and landmarks in the NeuFace-dataset and train the dedicated face motion prior, called HuMoR-Face. We train three motion prior models (HuMoR-Face) with different training datasets, *i.e.*, VOCASET (Cudeiro et al., 2019), NeuFace$_{\mathrm{MEAD}}$, and NeuFace$_{\mathrm{VoxCeleb2}}$. Please refer to appendix and HuMoR (Rempe et al., 2021) for the details.

**Long-term face motion generation.** We evaluate the validity and generative power of the learned motion prior by generating long-term 3D face motion sequences (10.0$s$). Long-term motions are generated by auto-regressive sampling from the learned prior, given only a starting frame as the condition (see Fig. 9). VOCASET provides small-scale, in-the-lab captured meshes, thus limited in motion naturalness and facial diversity. Accordingly, the HuMoR-Face trained with VOCASET fails to learn a valid human facial motion prior and generates unnatural motion. Using only the subset, NeuFace$_{\mathrm{MEAD}}$, the long-term stability of head motion has significantly enhanced. Please refer to the supplementary video for motion comparisons.

We quantitatively evaluate HuMoR-Face models using two metrics: motion Fréchet distance (FD) (Ng et al., 2022) and average pairwise distance (APD) (Aliakbarian et al., 2020; Rempe et al., 2021). FD measures the *naturalness* like the FID score (Heusel et al., 2017) and APD measures the *diversity* of generated motions. For APD, we generate 50 long-term motions from the same initial state and compute the mean landmark distance between all pairs of samples. HuMoR-Face models trained with large-scale and diverse motions, *i.e.*, the NeuFace-dataset, show superior performance in *naturalness* and *diversity* (see Table 4). Specifically, the HuMoR-Face trained with NeuFace$_{\mathrm{VoxCeleb2}}$ shows substantial enhancement on APD. APD is not reported for the HuMoR-Face trained with VOCASET, since the model fails to generate realistic motion. We attribute such high-quality motion prior to the benefit of the NeuFace-dataset: *large-scale* facial motion annotations. Further, exploiting diverse *in-the-wild*, *dynamic*, and *natural* motion annotation from NeuFace$_{\mathrm{VoxCeleb2}}$ helps HuMoR-Face learn real-world motion prior and surprisingly generate much diverse and dynamic motions.

Table 5: **Improving the face reconstruction accuracy.** (a) NeuFace-dataset helps the model reconstruct more occlusion robust and expressive 3D faces than the original model. Green and red dots denote visible and invisible 3D landmarks, respectively. (b) As a result, DECA$_{\text{NeuFace, 2D}}$, DECA$_{\text{NeuFace,3D}}$ achieve better 3D reconstruction accuracy than DECA$_{\text{original}}$.

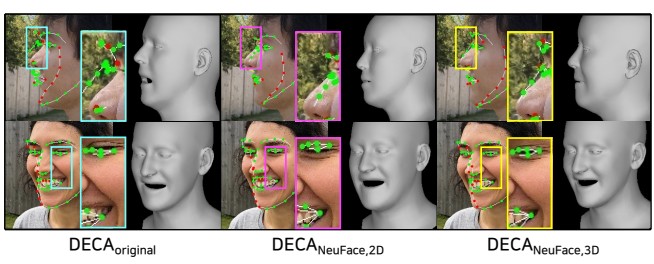

DECA$_{\text{original}}$      DECA$_{\text{NeuFace,2D}}$      DECA$_{\text{NeuFace,3D}}$

(a)

| Model | Test-opt | Error [mm] (↓) | | |
|---|---|---|---|---|
| | | Median | Mean | Std |
| 3DMM-CNN (Tuan Tran et al., 2017) CVPR 2017 | | 1.84 | 2.33 | 2.05 |
| PRNet (Feng et al., 2018) ECCV 2018 | | 1.50 | 1.98 | 1.88 |
| RingNet (Sanyal et al., 2019b) CVPR 2019 | | 1.21 | 1.54 | 1.31 |
| MGCNet (Shang et al., 2020) ECCV 2020 | | 1.31 | 1.87 | 2.63 |
| 3DDFA-V2 (Guo et al., 2020) ECCV 2020 | ✓ | 1.23 | 1.57 | 1.39 |
| DenseLandmarks (Wood et al., 2022) ECCV 2022 | ✓ | 1.02 | 1.28 | 1.08 |
| MICA (Zielonka et al., 2022) ECCV 2022 | ✓ | 0.90 | 1.11 | 0.92 |
| DECA$_{\text{original}}$ (Feng et al., 2021) SIGGRAPH 2021 | | 1.18 | 1.46 | 1.25 |
| DECA$_{\text{NeuFace,2D}}$ (Ours) | | 1.15 | 1.44 | 1.26 |
| DECA$_{\text{NeuFace,3D}}$ (Ours) | | **1.11** | **1.38** | **1.19** |

(b)

## 5.2 Improving the 3D reconstruction accuracy

Due to the absence of large-scale 3D face video datasets, existing face mesh regressor models utilize limited visual cues, such as 2D landmarks or segmentations. Thus, we utilize the NeuFace-dataset to add direct 3D supervision to enhance the performance of such a model.

**3D supervision with the NeuFace-dataset.** We implement the auxiliary 3D supervision as conventional 3D vertex and landmark losses (Kolotouros et al., 2019; Cho et al., 2022; Lin et al., 2021b;a). Given regressed and our annotated mesh vertices, $\mathbf{M}, \hat{\mathbf{M}} \in \mathbb{R}^{N_M \times 3}$, and regressed and our annotated 3D landmarks, $\mathbf{J}, \hat{\mathbf{J}} \in \mathbb{R}^{N_J \times 3}$, the auxiliary 3D losses are computed as: $\mathcal{L}_{\text{3D}}^{\text{M}} = \frac{1}{N_M} \|\mathbf{M} - \hat{\mathbf{M}}\|_2$, $\mathcal{L}_{\text{3D}}^{\text{J}} = \frac{1}{N_J} \|\mathbf{J} - \hat{\mathbf{J}}\|_2$, where $N_M$, $N_J$ is the number of mesh vertices and landmarks, respectively.

**Enhancement on 3D reconstruction accuracy.** By fine-tuning DECA (Feng et al., 2021) using the images of MEAD (Wang et al., 2020), VoxCeleb2 (Chung et al., 2018) and CelebV-HQ (Zhu et al., 2022), with and without our 3D supervision, we obtain DECA$_{\text{NeuFace,3D}}$ and DECA$_{\text{NeuFace,2D}}$. Following the evaluation protocol of the NoW benchmark (Sanyal et al., 2019a), we reconstruct 3D faces for the provided images via each model and report the 3D reconstruction errors. In Table 5, our DECA$_{\text{NeuFace,3D}}$ shows lower 3D reconstruction error than DECA$_{\text{original}}$ and DECA$_{\text{NeuFace,2D}}$.

## 6 Discussion, Limitation and Conclusion

We develop NeuFace, an optimization for generating accurate and spatio-temporally consistent 3D face mesh pseudo-labels on videos with a provable optimal guarantee. Moreover, with the technique, we build the NeuFace-dataset, a large-scale 3D face meshes paired with in-the-wild 2D videos.

We demonstrate the potential of the diversity and naturalness of our NeuFace-dataset as a training dataset to learn generative 3D facial motion prior. Also, we improve the reconstruction accuracy of a *de-facto* standard 3D face reconstruction model using our dataset. Note that the focus of our NeuFace optimization is to build the 3D face dataset for videos. The experiment for the static face reconstruction in Table 5 was conducted to demonstrate the potential application of our method and dataset. Given our emphasis on video face data, our model may not obtain the best accuracy for static face reconstruction. However, it is important to note that our method still shows a marked improvement over existing data. This underscores the effectiveness of the NeuFace-dataset in enhancing 3D face reconstruction. We expect NeuFace to open up new opportunities by providing large-scale, real-world 3D face video data, the NeuFace-dataset, as a reliable data curation method. The failure cases of NeuFace optimization could occur when the 2D video contains extreme degradations, e.g., motion blur, low resolution, extremely (>50%) occluded, so that the 2D keypoint detection fails. Please note that when we construct the NeuFace-dataset, we tackle these cases with automatic filtering followed by human verification (will be discussed in appendix Sec. A), which guarantees the reliability of the dataset. Since 2D landmark human annotations are relatively cheaper than any other signals, we think using better 2D landmarks can mitigate this limitation.

## Acknowledgment

This research was supported by a grant from KRAFTON AI, and also partially supported by Institute of Information & communications Technology Planning & Evaluation (IITP) grant funded by the Korea government(MSIT) (No.RS-2023-00225630, Development of Artificial Intelligence for Text-based 3D Movie Generation; No.RS-2022-II220290, Visual Intelligence for Space-Time Understanding and Generation based on Multi-layered Visual Common Sense; and No.RS-2022-II220124, Development of Artificial Intelligence Technology for Self-Improving Competency-Aware Learning Capabilities).

## Ethics Statement

For face reconstruction tasks and datasets, the diversity of race or ethnicity, gender, appearance, and actions is an important topic to discuss (Wang et al., 2019; Zhu et al., 2022). Existing 3D face video datasets (Zielonka et al., 2022; Ranjan et al., 2018; Cudeiro et al., 2019) typically have limited diversity regarding ethnicity, gender, appearance, and actions. Such 3D face datasets rarely provide video pairs, but with artificial facial markers attached to human faces and a small set of identities. On the other hand, our NeuFace-dataset mitigates such issues since our dataset is acquired on top of large-scale in-the-wild face video datasets, which typically rely on internet videos. Such video datasets are diverse in terms of ethnicity, gender, facial appearances, and actions when compared to the small/medium-scale 3D facial motion capture datasets. Since our dataset is acquired based on the existing public video datasets (Wang et al., 2020; Chung et al., 2018; Zhu et al., 2022), all the rights, licenses, and permissions follow the original datasets. Moreover, we will release the NeuFace-dataset by providing the reconstructed 3DMM parameters without the actual facial video frames. NeuFace-dataset does not contain identity-specific metadata and facial texture maps. Nonetheless, per-identity shape coefficients can give a rough guide about human facial shape. Thus, we release our dataset for research purposes only.

## Reproducibility Statement

We will make our code and data accessible to the public.

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

## Appendix

We present additional details, analysis, results, and experiments that are not included in the main paper due to the space limit. Also, the attached video explains and demonstrates the main idea of NeuFace and video samples for the NeuFace-dataset.

## A    Implementation details

We provide detailed configurations for implementations and experiments in the main paper.

**NeuFace optimization details.**    NeuFace optimization is composed of a neural network $\Phi_{\mathbf{w}}$ and the optimizing part. For the network $\Phi_{\mathbf{w}}$, we use a pre-trained DECA (Feng et al., 2021) or a pre-trained EMOCA (Danecek et al., 2022) encoder network. Overall optimization takes about 8 min. for 7 views, $\sim$120 frames of videos, and about 2.5 min. for 1 view, $\sim$120 frames of videos.

**NeuFace-dataset acquisition.**    We provide reliable 3D face mesh annotations for large-scale face video datasets: MEAD (Wang et al., 2020), VoxCeleb2 (Chung et al., 2018), and CelebV-HQ (Zhu et al., 2022). We optimize our full objective (Eq. (1)) to acquire FLAME meshes for the datasets with a multi-view camera setup, *e.g.*, MEAD. Otherwise, we optimize (Eq. (1)) with $\lambda_{\text{view}}{=}0$. We automatically discard the sequences if the optimization yields out-of-distribution shape parameters, *i.e.*, the L2-norm of shape parameters deviates largely from the pre-built distribution (Li et al., 2017), $\|\boldsymbol{\beta}\|_2{>}1.0$, or if the 2D landmark detector (Bulat & Tzimiropoulos, 2017) fails to capture the faces. After subsequent human verification, the NeuFace-dataset achieves $<0.1\%$ of failure rate on upon criteria, supporting the reliable quality of our dataset.

NeuFace dataset includes 24 diverse emotion/expression categories. Compared to indoor motion capture datasets (BIWI 3D, COMA, VOCASET), our dataset has a greater variety of emotion/expression categories. Moreover, our dataset contains 3D faces in natural talking scenarios, resulting in higher diversity within these categories. NeuFace-dataset contains at least 40 different facial attributes, including hairstyles and accessories such as glasses. While traditional motion capture datasets do not disclose the number of facial attributes, their facial attribute diversity is incomparably limited since they contain a small number of identities, and indoor environments. Unlike traditional motion capture datasets, which are recorded in controlled indoor environments with fixed camera views and lighting, the NeuFace-dataset uses in-the-wild YouTube videos, providing a much greater diversity of backgrounds.

**Facial motion prior.**    As one of our dataset's prominent applications, we introduced the learning of 3D facial motion prior, called HuMoR-Face (Sec.5.1 in the main paper). We first pre-process 3D face meshes in the NeuFace-dataset. We compute root orientation, face pose angles, 3D landmark positions, and their velocities, respectively. Then, we represent the state of a moving human face as $\mathbf{x} = [\boldsymbol{\phi}, \dot{\boldsymbol{\phi}}, \boldsymbol{\theta}, \mathbf{J}, \dot{\mathbf{J}}]$, where $\boldsymbol{\phi}$, $\dot{\boldsymbol{\phi}}$ denotes head root orientation and its velocity, $\boldsymbol{\theta}$ denotes the FLAME face pose parameters, and $\mathbf{J}$, $\dot{\mathbf{J}}$ denotes facial joint and its velocity, respectively.

The generative facial motion prior is trained to predict the facial motion state, $\mathbf{x}_{t+1}$, given the current state $\mathbf{x}_t$ as a condition. We consider the NeuFace holdout test split as the real motion distribution and compute the FD for the generated motions. We do not consider VOCASET as the real motion distribution for computing FD. It is limited in diversity and naturalness, which contradicts FD's purpose of measuring the naturalness of generated motions. Our NeuFace holdout test split is much larger and more diverse than VOCASET.

**Fine-tuning face mesh regressor.**    As our dataset's another application, we improve the accuracy of the pre-trained DECA model with our NeuFace-dataset and its 3D annotations. Specifically, we fine-tune the pre-trained DECA parameters with our NeuFace-dataset and the auxiliary 3D supervisions proposed in the main paper (L795-797). During fine-tuning, we use an adjusted learning rate, $1 \times 10^{-5}$, which is ten times smaller than training DECA from scratch. Note that there exist the DECA-coarse model and the DECA-detail model. Unfortunately, there are known issues in reproducing DECA-detail due to the absence of VGGFace2 and the training recipe (DECA GitHub issues: bit.ly/3jj2psn, bit.ly/3HIiVf0).

Table S1: **Average vs. Median operation.** we conduct an ablation study on a subset of the MEAD dataset, using a median operation instead of the average for the temporal consistency loss.

| Loss config. | CVD $\downarrow$ | NME$\downarrow$ |
|---|---|---|
| Temporal loss (Average) | **0.0981** | **3.03** |
| Temporal loss (Median) | 0.0985 | 3.09 |

Table S2: **Ablation on the design choices.** (a) Optimizing photometric loss ($\mathcal{L}_{\text{photo}}$) with the NeuFace optimization results in performance degradation. (b) The identity code ($\boldsymbol{\beta}$) gradually converges over the iterations, although we do not manually force shared identity regularization.

| Loss config. | CVD $\downarrow$ | MSI$_{\text{3D}}^{\text{L}}$ $\uparrow$ | MSI$_{\text{3D}}^{\text{V}}$ $\uparrow$ | NME$\downarrow$ |
|---|---|---|---|---|
| NeuFace | **0.103** | **0.206** | **0.305** | **2.58** |
| NeuFace$+2\cdot\mathcal{L}_{\text{photo}}$ | 0.106 | 0.205 | 0.299 | 2.62 |
| NeuFace$+5\cdot\mathcal{L}_{\text{photo}}$ | 0.112 | 0.195 | 0.282 | 3.69 |

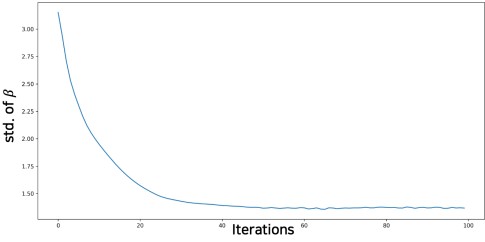

(a) Ablation on the photometric loss          (b) Std. of $\boldsymbol{\beta}$ over the iterations

# B    Ablation on the design choices

In our proposed NeuFace optimization, the temporal consistency loss employs a temporal moving average to estimate latent target meshes that represent temporally smooth heads. To assess the effectiveness of the temporal moving average operation, we conduct an ablation study using a median operation instead of the average on a subset of the MEAD dataset. In Table S1, both methods—temporal moving average and median operation in the temporal window—demonstrated comparable performance in terms of NME and CVD, with no significant difference in the optimization results. This finding suggests that the choice of operation has a negligible impact on the optimization process.

Although our proposed NeuFace optimization utilizes a simple combination of losses, we find that it is surprisingly effective for constructing accurate pseudo 3D face data. For example, we observe that the use of photometric loss ($\mathcal{L}_{photo}$) has a negligible effect, as demonstrated in Table S2 (a). In fact, $\mathcal{L}_{photo}$ tends to degrade performance, with larger values of this loss leading to greater degradation. We postulate that this might be due to the impact of self-shadows and non-Lambertian reflection caused by lighting and noise in video data, which could interfere with robust optimization. Interestingly, we also reveal that without explicit regularization for shared identities across frames in videos, the identity codes tend to converge automatically, as shown in Table S2 (b). This finding suggests that adding shared identity regularization may be unnecessary. Despite the effectiveness of our proposed NeuFace optimization, we believe that exploring additional losses to further refine the optimization process could be a worthwhile future direction.

We proposed an EM-style optimization for our NeuFace optimization. In each optimization step $t$, we compute latent target variables $\hat{\mathbf{M}}_f^t$ and $\hat{\mathbf{q}}_{f,v}^t$ at E-step using multi-view bootstrapping or temporal smoothing. The latent target variables serve as the supervision for spatio-temporal consistency losses at M-step. Our EM-style losses are as follows (Eq. (3) and Eq. (4) in the main):

$$\mathcal{L}_{\text{temporal,ours}} = \sum_{f=1,v=1}^{N_F,N_V}\|\mathbf{q}_{f,v}^t - \hat{\mathbf{q}}_{f,v}^t\|_2, \qquad \mathcal{L}_{\text{multiview,ours}} = \sum_{f=1,v=1}^{N_F,N_V}\|\mathbf{M}_{f,v}^t - \hat{\mathbf{M}}_f^t\|_1,$$

where $f$, $v$, $N_F$, and $N_V$ denote the temporal frame index, view index, total number of frames, and total number of views, respectively. Also, $\mathbf{M}, \mathbf{q}$ denotes the mesh vertices and head rotation quaternion, respectively. Please refer to Sec. 3.2 in the main draft for more details.

The naïve, non EM-style way to implement our optimization is as follows:

1. $\mathcal{L}_{\text{temporal,naive}}$ : Iterate through all the head rotation quaternion and compute frame-wise, Markov chain style head rotation error, without introducing latent target variable.

2. $\mathcal{L}_{\text{multiview,naive}}$ : Iterate through all the multi-view meshes in the same time frame and compute view-wise vertex error, without introducing latent target variable.

More formally, the losses for non EM-style optimization are as follows:

$$\mathcal{L}_{\text{temporal,naïve}} = \sum_{v=1}^{N_V} \sum_{f=1}^{N_F-1} \left\| \mathbf{q}_{f,v}^t - \mathbf{q}_{f+1,v}^t \right\|_2, \qquad \mathcal{L}_{\text{multiview,naïve}} = \sum_{f=1}^{N_F} \sum_{i=1}^{N_V-1} \sum_{j=i+1}^{N_V} \left\| \mathbf{M}_{f,i}^t - \mathbf{M}_{f,j}^t \right\|_1.$$

We conduct an ablation experiment that compares each optimization performance on a subset of the MEAD dataset. In Table S3, our EM-style optimization achieves better mesh quality in terms of temporal smoothness, cross-view vertex consistency, and landmark accuracy compared to the non EM-style optimization baseline.

Table S3: Ablation on EM-style optimization

| Optim. style | $\mathrm{MSI}_{\mathrm{3D}}^{\mathrm{L}} \uparrow$ | $\mathrm{MSI}_{\mathrm{3D}}^{\mathrm{V}} \uparrow$ | CVD $\downarrow$ | NME $\downarrow$ |
|---|---|---|---|---|
| Non EM-style optim. | 0.208 | 0.333 | 0.233 | 4.26 |
| **EM-style optim. (ours)** | **0.284** | **0.462** | **0.201** | **3.36** |

## C  NeuFace optimization with EMOCA

Recall that we can replace the neural parameterization of face meshes with another neural model. Specifically, we use EMOCA (Danecek et al., 2022), which is built upon DECA with an additional expression encoder. We change the neural network from DECA to EMOCA and optimize over it with our spatio-temporal and landmark losses. In Table 2 and Sec. 4 in the main paper, we discussed about the quantitative quality of NeuFace-E-dataset.

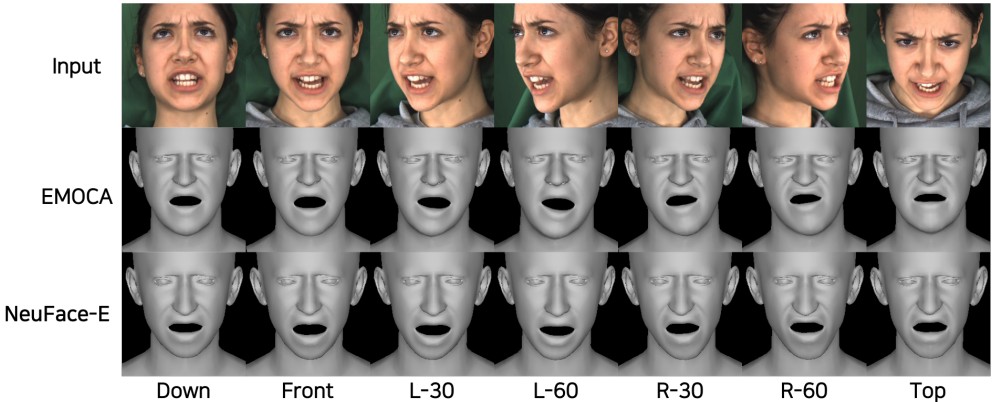

Figure S1: **Multi-view consistency: EMOCA vs. NeuFace-E.** We visualize the meshes obtained by EMOCA (Danecek et al., 2022) and NeuFace-E. By aligning the meshes to face the same direction, we can clearly notice that NeuFace-E obtains multi-view consistent meshes compared to EMOCA.

Qualitatively, we visualize the rendered meshes over different views as in Fig. S1. NeuFace-E-dataset ($3rd$ row) contains multi-view consistent meshes over views compared to the meshes obtained by EMOCA inference ($2nd$ row). Specifically, EMOCA produces huge discrepancies in mouth area across the views, while our method is more consistent. Moreover, our method reconstructs more accurate meshes, especially for the shape of the nose and face contour.

# D  NeuFace vs. Video 3D Face Tracking Method

We verify the favorable quality of NeuFace-datasets by comparing it with the meshes obtained by the state-of-the-art method, MICA with a tracker (Zielonka et al., 2022) (MICA+T), among the existing methods (Cao et al., 2013; 2015; Thies et al., 2016b; Zielonka et al., 2022). MICA+T jointly optimizes 3DMM, cameras, and textures with landmark and photometric losses, and statistic regularizers. In Table S4, NeuFace performs better than MICA+T in CVD & NME with comparable MSI. Also, faster optimization makes NeuFace preferable when annotating large-scale videos.

Table S4: Evaluation on MEAD: MICA vs. NeuFace

|  | CVD $\downarrow$ | MSI$_{3D}$ $\uparrow$ | NME $\downarrow$ | Optim. time (7 views) |
|---|---|---|---|---|
| MICA+T | 0.049 | **0.349** | 2.98 | 60 min. |
| **NeuFace-D-dataset** | **0.0094** | 0.277 | **2.58** | **8 min.** |

# E  Analysis on NeuFace

In this section, we introduce and validate our design choices for NeuFace optimization, through analysis. Specifically, we build a strong baseline and support our choice of "re-parameterized" face mesh optimization method for NeuFace in Sec. E.1. Next, we provide a proof sketch of the provable global minima convergence of NeuFace optimization in Sec. E.2.

## E.1  FLAME fitting vs. NeuFace optimization

Recall that NeuFace re-parameterizes the 3DMM, *i.e.*, FLAME (Li et al., 2017) to the neural parameters (represented as DECA (Feng et al., 2021)), then optimizes over them to obtain accurate 3D face meshes for videos. Following the prior arts in the parametric human body reconstruction literature (Bogo et al., 2016; Pavlakos et al., 2019; Kolotouros et al., 2019), there exists a simple method to optimize the parametric model; 3DMM parameter fitting. Thus, we implement FLAME fitting as a solid baseline and compare the quantitative and qualitative results with NeuFace optimization to analyze and support our choice of neural re-parameterization.

**Details of baseline FLAME fitting.**  Given the initial FLAME and camera parameters, $[\mathbf{\Theta_b}, \mathbf{p_b}] = [\mathbf{r_b}, \boldsymbol{\theta_b}, \boldsymbol{\beta_b}, \boldsymbol{\psi_b}, \mathbf{p_b}]$, we implement the direct FLAME optimization as:

$$[\mathbf{\Theta_b^*}, \mathbf{p_b^*}] = \underset{\mathbf{\Theta_b}, \mathbf{p_b}}{\arg\min}\ \mathcal{L}_{2D} + \lambda_{\text{temp}}\mathcal{L}_{\text{temporal}} + \lambda_{\text{view}}\mathcal{L}_{\text{multiview}} + \lambda_{\mathbf{r}}\mathcal{L}_{\mathbf{r}} + \lambda_{\boldsymbol{\theta}}\mathcal{L}_{\boldsymbol{\theta}} + \lambda_{\boldsymbol{\beta}}\mathcal{L}_{\boldsymbol{\beta}} + \lambda_{\boldsymbol{\psi}}\mathcal{L}_{\boldsymbol{\psi}},$$

where the losses $\mathcal{L}_{2D}$, $\mathcal{L}_{\text{temporal}}$, and $\mathcal{L}_{\text{multiview}}$ are identical to the losses discussed in the main paper (Eqs. 2,3,4). We can obtain the initial FLAME parameters for the optimization in two ways: (1) initialize from mean parameters and (2) initialize from pre-trained DECA (Feng et al., 2021) predictions. We empirically found that initialization with mean FLAME parameters frequently fails when the input images contain extreme head poses. Thus, we choose to initialize FLAME parameters from the pre-trained DECA predictions, thus providing a plausible initialization for a fair comparison. Also, following the convention (Li et al., 2017), we optimize FLAME parameters in a coarse-to-fine manner. For the earlier stage, we fix FLAME parameters that control local details, *i.e.*, $\boldsymbol{\theta_b}$, $\boldsymbol{\beta_b}$, and $\boldsymbol{\psi_b}$, and optimize the global head orientation, $\mathbf{r_b}$, and camera parameters $\mathbf{p_b}$. Then we fix camera parameters and optimize other FLAME parameters jointly at a later stage to fit the local details.

Since we initialize FLAME parameters and camera parameters from the pre-trained DECA predictions, *i.e.*, initial $[\mathbf{\Theta_b}, \mathbf{p_b}]$ in Eq. (5). Accordingly, the meshes obtained by the FLAME fitting achieve better spatio-temporal consistency and 2D landmark accuracy than the meshes obtained by a pre-trained DECA without any post-processing.

**Qualitative result.**  In Fig. S2(a), NeuFace-dataset contains much expressive and image-aligned meshes, *e.g.*, wrinkles and face boundaries. On the other hand, the meshes obtained by the direct FLAME optimization show

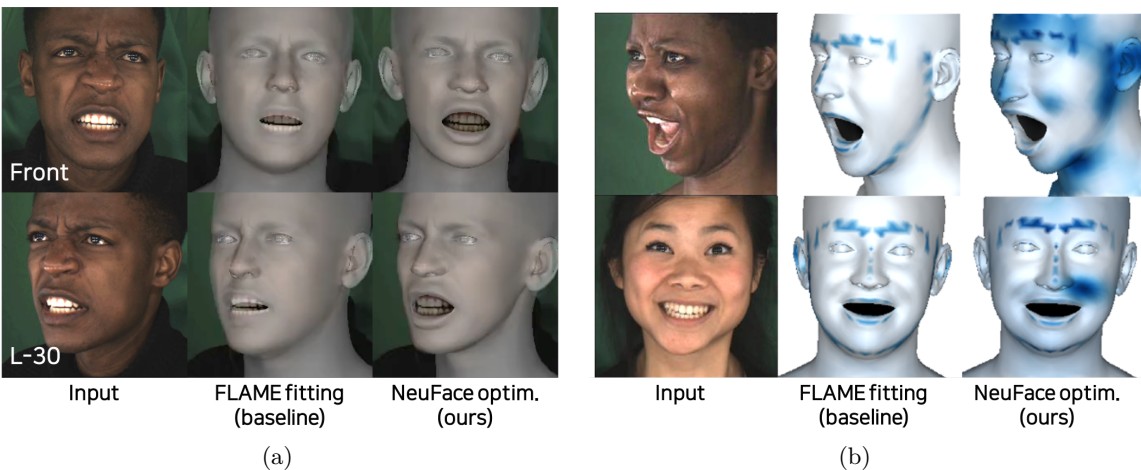

Figure S2: **FLAME fitting vs. NeuFace optimization.** (a) NeuFace optimization obtains much expressive and pixel-level aligned meshes than the baseline FLAME fitting. (b) We observe NeuFace optimization induces richer and data-dependent gradients compared to the sparse gradients of baseline.

mean shape-biased 3D faces. We explain such results in terms of the data-independency of the direct FLAME optimization. The baseline method requires several regularization terms based on the prior, pre-built from external 3D datasets, which is data-independent (Eq. (5)). Such data-independent regularization encourages the optimized FLAME parameters to stay close to the mean of parameter distributions, regardless of the facial characteristics on input images. Balancing such regularization terms with other losses is cumbersome and prone to obtain mean shape faces.

In contrast, recall that NeuFace re-parameterizes the FLAME parameters as the pre-trained neural network, such as DECA (Feng et al., 2021). Such re-parameterization allows NeuFace to update face meshes in an input-image-conditioned manner, called data-dependent mesh update (Sec. 3.3 in the main paper). Figure S2(b) visualizes the data-dependent gradient of our optimization. While the baseline shows similar gradient patterns throughout three input images, NeuFace optimization produces diverse gradient maps according to input images. Such rich and data-dependent gradients consider high-dimensional visual features induced from the input RGB values in the optimization process, yielding accurate and expressive 3D faces.

### E.2 Proposition 1: Convergence Property to Global Minima

Proposition 1 in this work is a straightforward variation of the main result of Allen-Zhu *et al.* (Allen-Zhu et al., 2019). Before providing the proof sketch of Proposition 1. We describe the assumptions needed to prove Proposition 1. For simplicity, we consider a simple $l_2$-based regression loss and a $L$-layer fully connected ReLU network $\Phi_{\overrightarrow{\mathbf{W}}}(\cdot)$ having a uniform weight width size of $m$. For the training data consisting of vector pairs $\{(x_i, y_i^*)\}_{i\in[n]}$, the network has the batched input of $\mathbf{x}=\{x_i\}_{i\in[n]}$, where $n$ is the batch size, $x_i\in[0,1]^{d_{in}}$, and $y_i^* \in \mathbb{R}^{d_{out}}$, the output of $\Phi_{\overrightarrow{\mathbf{W}}}(x) \in \mathbb{R}^{d_{out}}$, and the weights $\overrightarrow{\mathbf{W}} = (\mathbf{W}_1\in\mathbb{R}^{m\times d_{in}}, \mathbf{W}_{\{2,\cdots,L-1\}}\in\mathbb{R}^{m\times m}, \mathbf{W}_L\in\mathbb{R}^{d_{out}\times m})$.[4]

**Assumption 1.** *Without loss of generality, $\forall i, \|x_i\| = 1$ and $\|y_i^*\| \leq O(1)$.*

**Assumption 2.** *The pretrained neural network weights $\overrightarrow{\mathbf{W}}^{(0)}$ are assumed to be started from the values distributed normally,* i.e., *considered as a sample instance from a Gaussian distribution. Specifically, $[\mathbf{W}_l^{(0)}]_{i,j} \sim \mathcal{N}(0, 2/\mathtt{row}[\mathbf{W}_l])$ for $l \in \{1,\cdots,L-1\}$ and $[\mathbf{W}_L^{(0)}]_{i,j} \sim \mathcal{N}(0, 1/\mathtt{row}[\mathbf{W}_L])$ for every $(i,j)$, where the operator $\mathtt{row}[\cdot]$ returns the row size of the input matrix.*

Assumption 2 appears to be restrictive by those standard deviations, but it is not. The assumptions cover a fairly broad range of weight distribution scenarios. For larger standard deviations, we can always set a small norm for $x$'s in Assumption 1 without loss of generality, and vice versa.

---

[4]We can consider that the biases are included in $\{\mathbf{W}\}$ without loss of generality.

Under these assumptions, we restate Proposition 1 in the main paper.

**Proposition 1** (Global Convergence). *For any $\epsilon \in (0, 1]$, $\delta \in \left(0, O(\frac{1}{L})\right]$, given an input data $\{\mathbf{x}, \mathbf{y}\}$ and the neural network $\Phi_{\overrightarrow{\mathbf{W}}}(\cdot)$ over-parameterized such that $m \geq \Omega\left(\texttt{poly}(n, L, \delta^{-1}) \cdot d_{out}\right)$, consider optimizing the non-convex regression problem:* $\arg\min_{\overrightarrow{\mathbf{W}}} \mathcal{L}(\overrightarrow{\mathbf{W}}) = \frac{1}{2}\|\Phi_{\overrightarrow{\mathbf{W}}}(\mathbf{x}) - \mathbf{y}\|_2^2$.
*Under the above assumptions, with high probability, the gradient descent algorithm with the learning rate $\rho = \Theta\left(\frac{d\delta}{\texttt{poly}(n, L)m}\right)$ finds a point $\overrightarrow{\mathbf{W}}^*$ such that $\mathcal{L}(\overrightarrow{\mathbf{W}}^*) \leq \epsilon$ in polynomial time.*

*Proof sketch.* We first introduce the following useful lemmas needed to prove the proposition.

**Lemma 1** (Theorem 3 in (Allen-Zhu et al., 2019)). *With probability at least $1 - e^{-\Omega(m/\Omega(\texttt{poly}(n, L, \delta^{-1})))}$, it satisfies for every $\ell \in [L]$, every $i \in [n]$, and every $\overrightarrow{\mathbf{W}}$ with $\|\overrightarrow{\mathbf{W}} - \overrightarrow{\mathbf{W}}^{(0)}\|_2 \leq \frac{1}{\texttt{poly}(n, L, \delta^{-1})}$, where $\|\overrightarrow{\mathbf{W}}\|_2 = \max_{l \in [L]} \|\mathbf{W}_l\|_2$,*

$$\Omega\left(\mathcal{L}(\overrightarrow{\mathbf{W}}) \cdot \frac{\delta m}{dn^2}\right) \leq \|\nabla\mathcal{L}(\overrightarrow{\mathbf{W}})\|_F^2 \leq O\left(\mathcal{L}(\overrightarrow{\mathbf{W}}) \cdot \frac{Lnm}{d}\right).$$

This lemma suggests that, when we are close to the starting point $\overrightarrow{\mathbf{W}}^{(0)}$ of the neural network, there is no saddle point or critical point of any order. Specifically, for example, given a fixed $\delta, d, n$ and $L$, when we have the same error $\mathcal{L}$ for two different neural networks with respective widths of $m_1$ and $m_2$, where $m_1 < m_2$, then the lower bound of the gradient with $m_2$ is larger than that of $m_1$ with a better chance. This means that neural networks with larger widths are likely to have a lower chance of local minima.

This hints that any local search (*e.g.*, gradient descent) does not suffer from any local minima or saddle points for larger $m$, which implies a more likely chance of avoiding local minima, *i.e.*, finding global minima. However, the local search does not guarantee to decrease the loss function yet.

With the favorable property of Lemma 1, if we have an additional guarantee of loss decrease with gradient descent, we can prove the convergence to global minima. To derive objective-decrease guarantee in optimization theory, a notion of smoothness is typically needed; thus, we introduce the following lemma.

**Lemma 2** (Theorem 4 of (Allen-Zhu et al., 2019)). *With probability at least $1 - e^{-\Omega(m/\texttt{poly}(L, \log m))}$, we have: for every $\overrightarrow{\mathbf{W}}^\dagger \in (\mathbb{R}^{m \times m})^L$ with $\|\overrightarrow{\mathbf{W}}^\dagger - \overrightarrow{\mathbf{W}}^{(0)}\|_2 \leq \frac{1}{\texttt{poly}(L, \log m)}$, and for every $\overrightarrow{\mathbf{W}}' \in (\mathbb{R}^{m \times m})^L$ with $\|\overrightarrow{\mathbf{W}}'\|_2 \leq \frac{1}{\texttt{poly}(L, \log m)}$, the following inequality holds*

$$\mathcal{L}(\overrightarrow{\mathbf{W}}^\dagger + \overrightarrow{\mathbf{W}}') \leq \mathcal{L}(\overrightarrow{\mathbf{W}}^\dagger) + \left\langle \nabla\mathcal{L}(\overrightarrow{\mathbf{W}}^\dagger), \overrightarrow{\mathbf{W}}' \right\rangle + O(\frac{nL^2m}{d})\|\overrightarrow{\mathbf{W}}'\|_2^2 + \frac{\texttt{poly}(L)\sqrt{nm\log m}}{\sqrt{d}} \cdot \|\overrightarrow{\mathbf{W}}'\|_2 \sqrt{\mathcal{L}(\overrightarrow{\mathbf{W}}^\dagger)}.$$

This lemma states the semi-smoothness property of the objective function $\mathcal{L}$ *w.r.t.* $\Phi_{\overrightarrow{\mathbf{W}}}(\cdot)$ to take into account non-smoothness introduced by ReLU activation in $\Phi_{\overrightarrow{\mathbf{W}}}(\cdot)$. The semi-smoothness looks similar to the Lipschitz smoothness except for the first order term $\|\overrightarrow{\mathbf{W}}'\|_2$. Interestingly, when we increase $m$, the increasing rate of the first order term is much slower than that of the second order term; thus, the second order term becomes dominant compared to the first order one, and the semi-smoothness approaches closer to the Lipschitz smoothness. This means that the neural network is smoother as $m$ goes larger.

Under the assumption that $\|\overrightarrow{\mathbf{W}}^{(t)} - \overrightarrow{\mathbf{W}}^{(0)}\|_F$ is small (will be verified later), the next step is to combine Lemma 2 with gradient descent to derive the loss-decrease guarantee. Denoting $\nabla_t = \nabla\mathcal{L}(\overrightarrow{\mathbf{W}}^{(t)})$, the gradient descent update rule is defined as: $\overrightarrow{\mathbf{W}}^{(t+1)} = \overrightarrow{\mathbf{W}}^{(t)} - \rho\nabla_t$ for a learning rate $\rho > 0$. Then, from Lemma 2, putting $\overrightarrow{\mathbf{W}}^{(t+1)} = \overrightarrow{\mathbf{W}}^\dagger + \overrightarrow{\mathbf{W}}'$ and $\overrightarrow{\mathbf{W}}^{(t)} = \overrightarrow{\mathbf{W}}^\dagger$, *i.e.*, $\overrightarrow{\mathbf{W}}' = -\rho\nabla_t$, we have

$$\mathcal{L}(\overrightarrow{\mathbf{W}}^{(t+1)}) \leq \mathcal{L}(\overrightarrow{\mathbf{W}}^{(t)}) - \rho\|\nabla_t\|_F^2 + \rho^2 C_1\|\nabla_t\|_2^2 + \rho C_2\|\nabla_t\|_2\sqrt{\mathcal{L}(\overrightarrow{\mathbf{W}}^{(t)})}$$

$$\text{(where } C_1 = O(\frac{nL^2m}{d}), C_2 = \frac{\texttt{poly}(L)\sqrt{nm\log m}}{\sqrt{d}})$$

$$\leq \mathcal{L}(\overrightarrow{\mathbf{W}}^{(t)}) - \rho\|\nabla_t\|_F^2 + \left(\rho^2 C_1 O\left(\frac{nm}{d}\right) + \rho C_2\sqrt{O\left(\frac{nm}{d}\right)}\right)\mathcal{L}(\overrightarrow{\mathbf{W}}^{(t)})$$

$$(\|\nabla_t\|_2^2 \leq \max_{l \in [L]} \|\nabla_{\mathbf{W}_l} \mathcal{L}(\overrightarrow{\mathbf{W}}^{(t)}))\|_F^2 \leq O\left(\tfrac{nm}{d}\right) \mathcal{L}(\overrightarrow{\mathbf{W}}^{(t)}) \text{ from the upper bound in Lemma 1)}$$

$$= (1 + C_3)\mathcal{L}(\overrightarrow{\mathbf{W}}^{(t)}) - \rho\|\nabla_t\|_F^2 \qquad (\text{where } C_3 = \rho^2 C_1 O\left(\tfrac{nm}{d}\right) + \rho C_2 \sqrt{O\left(\tfrac{nm}{d}\right)})$$

$$\leq \left(1 - \Omega\left(\frac{\rho\delta m}{dn^2}\right)\right) \mathcal{L}(\overrightarrow{\mathbf{W}}^{(t)})$$

(by the gradient lower bound from Lemma 1 and our choice of $\rho$, $e.g.$, $\rho = \Theta(\frac{d\delta}{n^4 L^2 m})$)

When we choose the parameters such that $\Omega\left(\frac{\rho\delta m}{dn^2}\right) \in (0,1)$, we have $\mathcal{L}(\overrightarrow{\mathbf{W}}^{(t+1)}) < \mathcal{L}(\overrightarrow{\mathbf{W}}^{(t)})$. In other words, there exists $T > 0$ such that $\mathcal{L}(\overrightarrow{\mathbf{W}}^{(T)}) \leq \epsilon$. Examples of convenient parameter choices of $m, \rho$, and $T$ in polynomial orders are suggested in Allen-Zhu et al. (2019) to hold $\Omega\left(\frac{\rho\delta m}{dn^2}\right) \in (0,1)$ and the small value of $\|\overrightarrow{\mathbf{W}}^{(t)} - \overrightarrow{\mathbf{W}}^{(0)}\|_F$ for every $t$. This concludes the proof sketch of finding a point $\overrightarrow{\mathbf{W}}^* = \overrightarrow{\mathbf{W}}^{(T)}$ such that $\mathcal{L}(\overrightarrow{\mathbf{W}}^*) \leq \epsilon$. $\qquad \square$

**Remark 1: Global optimality.** In Proposition 1, we can set $\epsilon$ arbitrarily small. With a very small $\epsilon$, it suggests that a converged point $\overrightarrow{\mathbf{W}}^*$ is a global minimum.

**Remark 2: The radius condition between the initial weights and updated one.** The spectral radius bounds for $\|\overrightarrow{\mathbf{W}} - \overrightarrow{\mathbf{W}}^{(0)}\|_F$ required in Lemmas 2 and 1 appear to be small, it is sufficiently large enough to completely change the output of the model, considering the large width of size $m$ and the standard deviation $\frac{1}{\sqrt{m}}$ of weight entries in Assumption 2.

**Remark 3: Other architectures.** Allen-Zhu et al. (2019); Du et al. (2019a) present the recipes to convert the $L$-layer fully connected networks to convolutional neural networks and to ResNet by sacrificing the complexity of proof. Thus, the conclusion of the provable guarantee does not change with such architectural changes. Thus, the architectures we experimented provably comply with the conclusion of Proposition 1 up to the choice of the parameters, $i.e.$, global convergence.

**Remark 4: Other losses.** In the above proof sketch, one of the important pieces is the semi-smoothness in Lemma 2. While we discuss only with the simple $l_2$ regression loss function, fortunately, the semi-smoothness already encompasses any choice of Lipschitz smooth cases for the loss functions. Thus, as long as the choice of the loss function is Lipschitz smooth, the replacement of the loss function does not alter the conclusion of Proposition 1 even for non-convex losses except the choice of parameters. This hints that our choice of the multi-task loss in Eq. (1) provably complies with the conclusion of Proposition 1 except the choice of the parameters.

**Remark 5: $L$ vs. $m$ for the over-parameterization.** For designing the over-parameterized architecture, one can control two different parameters $L$ and $m$.

Obviously, the high probability is achieved with larger $m$ rather than $L$, but more importantly, the local minima smoothing phenomenon suggested in the lower bound of Lemma 1 is independent to $L$.

## F    More dataset samples

We present more qualitative samples of NeuFace-dataset, with diverse identities and visual features (see Fig. S3). Since we cannot deliver expressive and temporally smooth facial motion in images, we strongly recommend seeing video visualizations for NeuFace-dataset, in the supplementary videos.

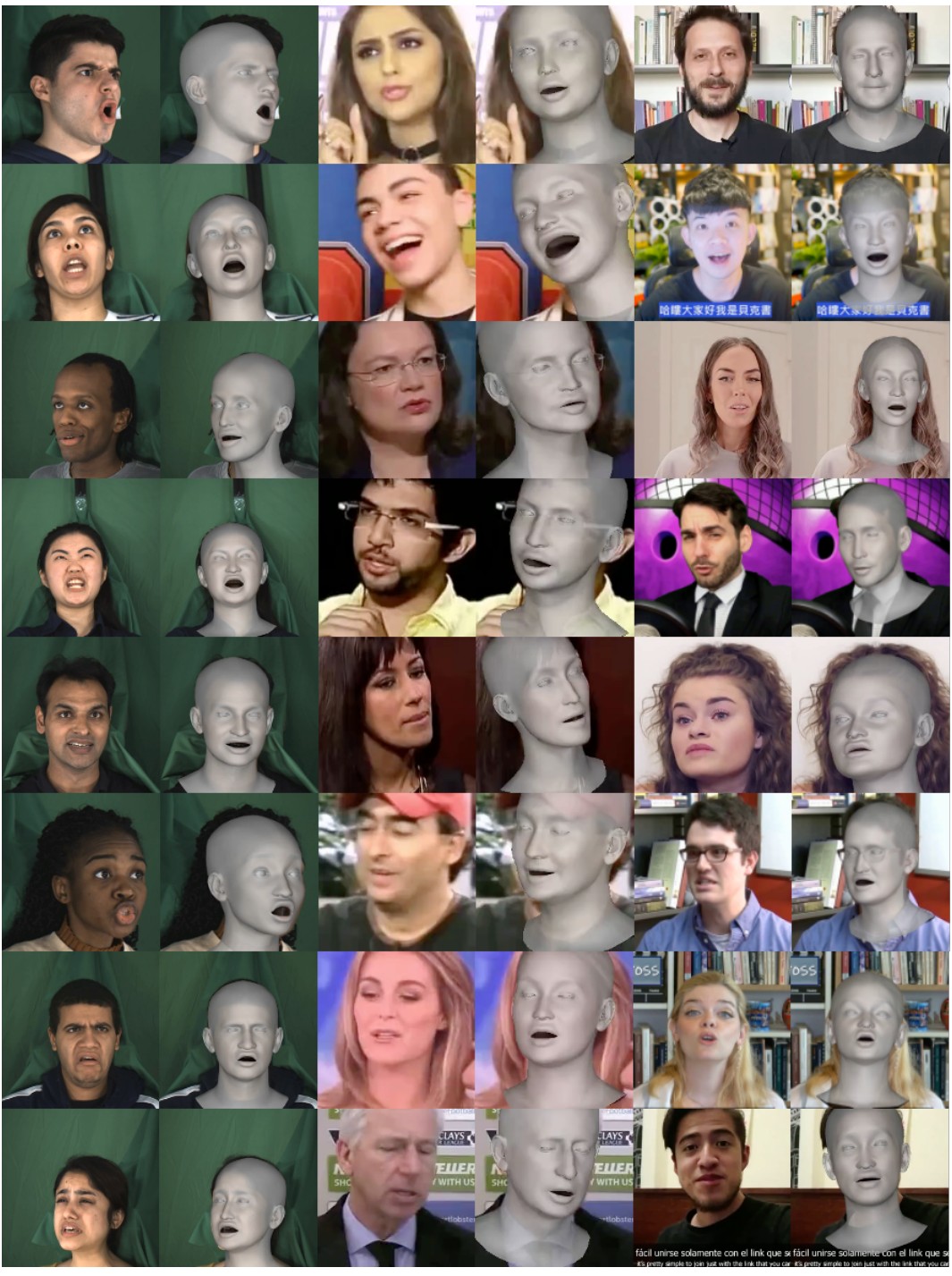

Figure S3: **NeuFace-dataset** is the large-scale 3D face video dataset containing 3DMM annotations for faces with diverse ethnicity, gender, emotions, and actions. See supplementary videos for the dynamic face visualizations.

