# OpenReview forum: "A Large-Scale 3D Face Mesh Video Dataset via Neural Re-parameterized Optimization"
_TMLR — Accepted by TMLR_

### Review · Reviewer_HkHi · 2024-03-08

**Summary Of Contributions:**

This work introduces a robust 3D face mesh pseudo annotation pipeline - NeuFace  - to annotate the 3D face mesh from 2D video. Based on this pipeline and extra manual verification, the largest 3D face mesh pseudo-labeled dataset - NeuFace-dataset - is constructed including the data captured in the wild setting. Furthermore, the performance improvement from this dataset has been demonstrated in two 3D face tasks.   Considering that the data-driven recipe is the most popular way in CV tasks, I believe this dataset could be helpful for 3D face tasks.

**Audience:**

Yes

**Broader Impact Concerns:**

Not necessary.

**Claims And Evidence:**

Yes

**Requested Changes:**

The detailed statistics on the dataset should be reported.
More ablations or explanations on the EM optimization choice could be illustrated to provide more insights.

**Strengths And Weaknesses:**

Pros:

- The overall annotation system is constructed based on prior works, including FLAME, DECA, and prior common regularization terms and losses. The temporal consistency loss and Multi-view consistency loss regularization terms are considered to improve the spatial-temporal consistency. Although this annotation system is mainly built on top of prior techniques, the introduced NeuFace optimization pipeline and the constructed dataset could be useful for academia and industry.

- NeuFace optimization could be a good choice for making pseudo annotation considering its effectiveness and efficiency evaluated in the supplementary materials.

- Fig. 3 is nice to visualize the Data-dependent gradient, which is the main motivation of the proposed NeuFace optimization method. It is explained that the over-parameterization method can achieve better global convergence. The proof sketch is also provided in the supplementary materials.  Fig.4 further proves the superiority of  NeuFace optimization over the prior methods.

- NeuFace-dataset achieves about 1,000 times larger number of sequences than existing facial motion capture datasets. Especially, VoxCeleb2 and CelebV-HQ are two datasets captured in-the-wild setting.

Cons:

- It is claimed that NeuFace-dataset is the largest 3D face mesh pseudo-labeled dataset in terms of the scale, naturalness, and diversity of facial attributes, emotions, and backgrounds.  However, the detailed statistics are not reported. I believe such statistics could make this paper more comprehensive.


- It is claimed that the Expectation-Maximization (EM) optimization style is chosen to ease the loss optimization. More ablations or explanations could be illustrated to provide more insights.

---

> ### Author Response · Authors · 2024-06-05
> **Response to Reviewer HkHi**
>
> We thank the reviewer for the time and the thorough review. By addressing the reviewer’s questions and comments, we could strengthen our paper. We address the concerns and the questions below and in the revision (please check pdf, highlighted in pink).
>
> Please let us know if our answers satisfy the reviewer’s concerns. We would be happy to provide further discussions and clarifications.
>
> **1. Detailed statistics of the NeuFace-dataset**
>
> Our NeuFace dataset is curated from videos in the MEAD, VoxCeleb2, and CelebV-HQ datasets, and follows their statistics. In addition to the number of sequences, identities, video lengths, and environments provided in Table 1 of the manuscript, we newly report additional statistics and discussion:
>
> - **Number of emotions/expressions:**
> NeuFace-dataset includes 24 diverse emotion/expression categories. Compared to indoor motion capture datasets (BIWI 3D, COMA, VOCASET), our dataset has a greater variety of emotion/expression categories. Moreover, our dataset contains 3D faces in natural talking scenarios, resulting in higher diversity within these categories.
> - **Number of facial attributes:**
> NeuFace-dataset contains at least 40 different facial attributes, including hairstyles and accessories such as glasses. While traditional motion capture datasets do not disclose the number of facial attributes, their facial attribute diversity is incomparably limited since they contain a small number of identities, and indoor environments.
> - **Diversity of background:**
> Unlike traditional motion capture datasets, which are recorded in controlled indoor environments with fixed camera views and lighting, the NeuFace-dataset uses in-the-wild YouTube videos, providing a much greater diversity of backgrounds.
>
> We have added the number of emotion/expression categories as a column in Table 1 and included the other details in the Appendix Sec A.
>
> **2. More ablations or explanations on the EM-style optimization choice**
>
> We proposed an EM-style optimization for our NeuFace optimization. In each optimization step $t$, we compute latent target variables $\hat{\mathbf{M}}\^{t}\_{f}$ and $\hat{\mathbf{q}}\^{t}\_{f,v}$ at E-step using multi-view bootstrapping or temporal smoothing. The latent target variables serve as the supervision for spatio-temporal consistency losses at M-step. Our EM-style losses are as follows (Eq.(3) and (4) in the main):
>
> $$
> \mathcal{L}\_\text{temporal,ours} = \sum\_{f=1,v=1}\^{N\_{F},N\_{V}}{\lVert \mathbf{q}\^{t}\_{f,v}-\mathbf{\hat{q}}\^{t}\_{f,v}\rVert}\_{2},\qquad
> \mathcal{L}\_{\text{multiview,ours}} = \sum\_{f=1, v=1}\^{N\_{F}, N\_{V}}{\lVert \mathbf{M}\^{t}\_{f,v}-\mathbf{\hat{M}}\^{t}\_f \rVert\_{1}},
> $$
>
> where $f, v, N\_F, N\_V$ denote the temporal frame index, view index, total number of frames and total number of views, respectively. Also, $\mathbf{M}, \mathbf{q}$ denotes the mesh vertices and head rotation quaternion, respectively. Please refer to the Sec.3.2 in the main draft for more details.
>
> The naive, non EM-style way to implement our optimization is as follows:
>
> - $\mathcal{L}\_\text{temporal,naive}$ : Iterate through all the head rotation quaternion and compute frame-wise, Markov chain style head rotation error, without introducing latent target variable.
> - $\mathcal{L}\_\text{multiview,naive}$ : Iterate through all the multi-view meshes in the same time frame and compute view-wise vertex error, without introducing latent target variable.
>
> $$
> \mathcal{L}\_\text{temporal,naive} =\sum\_{v=1}\^{N\_{V}}\sum\_{f=1}\^{N\_{F}-1} \lVert \mathbf{q}\^{t}\_{f,v} - \mathbf{q}\^{t}\_{f+1,v} \rVert\_2,\qquad
> \mathcal{L}\_\text{multiview,naive}= \sum\_{f=1}\^{N\_{F}}\sum\_{i=1}\^{N\_{V}-1}\sum\_{j=i+1}\^{N\_{V}} \lVert\mathbf{M}\^{t}\_{f,i}-\mathbf{M}\^{t}\_{f,j}\rVert\_{1}.
> $$
>
> We have newly conducted the ablation experiment that compares each optimization performance on the subset of MEAD dataset. The following table shows that our EM-style optimization helps achieve better mesh quality in terms of temporal smoothness, cross-view vertex consistency, and landmark accuracy, compared to the naive non EM-style optimization baseline.
>
> |  | $\text{MSI}\_{3D}\^{L} (\uparrow)$ | $\text{MSI}\_{3D}\^{V} (\uparrow)$ | $\text{CVD} (\downarrow)$ | $\text{NME}(\downarrow)$ |
> | ----- | --- | --- | --- | --- |
> | Non EM-style optim. | 0.208 | 0.333 | 0.233 | 4.26 |
> | **EM-style optim. (Ours)** | **0.284** | **0.462** | **0.201** | **3.36** |
>
> We have newly added this discussion in the Appendix Sec. B and Table S3.

---

### Review · Reviewer_BzRa · 2024-04-17

**Summary Of Contributions:**

This paper introduce a 3D face video dataset with pseudo ground-truth meshes, built on existing datasets.  To obtain the accurate face meshes,  the authors finetune a pre-trained  DECA / EMOCA  network with the proposed temporal and multi-vew consistency regularization.  Then the meshes are generated from face videos.   The proposed dataset can benefit the face reconstruction and face motion generation tasks.

**Audience:**

Yes

**Claims And Evidence:**

Yes

**Requested Changes:**

The authors are suggested to revise the methodolgy based on the weaknesses part.

More experiments and disscusions about data validation are  highly recommanded.

There is a duplicated sentence in page 5:  "We set the vertices as invisible"

**Strengths And Weaknesses:**

The paper addresses an important problem - the lack of a face video dataset with ground truth 3D geometry. The motivation for this work is valuable.

The proposed NeuFace optimization scheme leverages temporal and spatial face data to improve face model parameter estimation. This helps alleviate issues like temporal jittering and multi-view inconsistencies, leading to good results in these aspects.

The authors demonstrate the applications of such a dataset well.

However, the paper's claim of "re-parameterizing" the FLAME parameters is questionable. The authors appear to simply use a network to predict the original FLAME parameters from input images, which is not technically a re-parameterization.

The arguments made for why using a network to predict model parameters is better than direct regression are not fully convincing. The gradient of the loss function in Equation 6 does contain information from the input image, which means it is a form of "data-dependent" .

There are also concerns about the evaluation methodology. It is challenging to accurately evaluate 3D mesh accuracy in the dataset, and the proposed methods do not seem to be the best performing in the reported results (Table 3). This raises the question of whether this is the best approach for generating such a dataset, or if alternative methods like RGB-D face video capture could provide more reliable ground truth 3D geometry.

Overall, the paper addresses an important problem, but there are some technical and evaluation-related issues that need to be addressed further.

---

> ### Author Response · Authors · 2024-06-05
> **Response to Reviewer BzRa (Part 1/2)**
>
> We thank the reviewer for the time and the thorough review, which helped us improve our paper. We address the concerns and the questions below and in the revision (please check the updated pdf, highlighted in pink).
>
> Please let us know if our answers address the reviewer’s concerns. We would be happy to provide further discussions and clarifications.
>
> **1. The claim of "re-parameterizing" the FLAME parameters is questionable. Using a network to predict the original FLAME parameters from input images is not technically a re-parameterization.**
>
> As acknowledged by both Reviewers **D2De** and **HkHi**, we propose to re-parameterize the FLAME parameters, so that we can get the benefits of over-parameterization by optimizing network parameters instead of direct optimization of the FLAME parameters. Also, we would like to respectfully correct the reviewer’s misunderstanding that our NeuFace optimization is not fine-tuning, but a test-time optimization for a single test sample at an inference time, i.e., working in an over-fitting regime. The benefit of doing this was analyzed in the theory and Figs. 3 and 4, as acknowledged by Reviewer **HkHi**.
>
>
>
> **2. Why using a network to predict model parameters is better than direct regression are not fully convincing. The gradient of the loss function in Equation 6 does contain information from the input image, which means it is a form of "data-dependent".**
>
> As the reviewers **HkHi** and **D2De** acknowledged, the benefit of our re-parameterized optimization is to leverage data-dependent gradient, which helps obtaining dense gradients even to non-landmark facial parts, which are important for getting high-quality face meshes.
>
> (Please refer to Reviewer **D2De**’s mention, “It's encouraging to see the gradients to update regions like the cheeks which do not traditionally have 2D landmarks but are important for high-quality face meshes.”)
>
> Regarding the gradient of the loss function in Equation 6, Equation 6 update does not consider abundant image pixel-level information and mainly focuses only on the supervised area, i.e., sparse facial landmarks when back-propagated. We’d like to clarify that the loss $\mathcal{L}$ is computed using the detached and pre-computed sparse 2D landmarks. The 2D landmarks that we used for computing $\mathcal{L}$ are acquired by running a detached, off-the-shelf landmark detection model, FAN (mentioned in Sec.3.2), which only stores the sparse, facial landmark image coordinates.
>
> On the contrary, our NeuFace update in Equation 7 incorporates gradients related to pixel-level feature $\frac{\partial}{\partial\mathbf{w}^t}\Phi\_{\mathbf{w}^{t}}({\mathbf{I}})$, thus it is a form of “input data-dependent”, and obtain dense gradient when back-propagated. This phenomenon is shown in Fig. 3, Fig. S2 and Sec. E.1 in the manuscript.

---

> ### Author Response · Authors · 2024-06-05
> **Response to Reviewer BzRa (Part 2/2)**
>
> **3. Proposed methods do not seem to be the best performing in the reported results (Table 3). Raises question if alternative methods like RGB-D face video capture could provide more reliable ground truth 3D geometry.**
>
> The main focus of our proposed NeuFace optimization is to build a large-scale “3D face dataset for videos,” rather than static faces.
>
> The experiment for the “static” face reconstruction in Table 3 was conducted to demonstrate one of the potential applications of our method & dataset. Given our emphasis on video face data, our model may not obtain the best accuracy for static face reconstruction in this current form (there might be potential if we deploy more accurate component modules). However, it is important to note that our method still shows marked improvements over the existing datasets. This underscores the effectiveness of NeuFace-dataset in enhancing 3D face reconstruction. Also, we evaluated the validity of our proposed pseudo-data annotation method in Figure 4 and Table 2 in the main draft. Please consider the main points of our work.
>
> By the reviewer’s comment, we found that the relevant experiments on monocular 3D face reconstruction including Table 3 were more emphasized than other applications in the manuscript. Therefore, we have reordered Sec. 5.1 (Improving the 3D reconstruction accuracy) and Sec. 5.2 (Learning 3D human facial motion prior) in the manuscript to address this confusion. Additionally, we have added a discussion and limitations of this experiment (Sec 6 in the revision) to tone down this part.
>
> Regarding alternative methods using RGB-D video, 1) RGB-D sensors already have centimeter-level errors [C1,C2,C3], which is much worse accuracy than the average error (1.38mm) of our model, 2) achieving the scale of our dataset, which is based on YouTube videos, would be impractical with such RGB-D sensors (As Reviewer **HkHi** acknowledged, NeuFace-dataset is about 1000 times larger than existing datasets, underscoring the significant advantage and scalability of our approach), 3) a seminal work [C4] that improves a single RGB-D camera’s accuracy with the 3D face model to capture 3D faces reported an average 3D reconstruction error of 1.66mm. In contrast, the neural models we improved using NeuFace-dataset already have a lower average error (1.38mm) on the 3D benchmark. Thus, the existence of RGB-D does not harm our contribution.
>
> [C1] Azure Kinect DK depth camera ([https://learn.microsoft.com/en-us/azure/kinect-dk/hardware-specification](https://learn.microsoft.com/en-us/azure/kinect-dk/hardware-specification)) — 17mm depth error
>
> [C2] Intel realsense ([https://www.intelrealsense.com/compare-depth-cameras/](https://www.intelrealsense.com/compare-depth-cameras/)) — 20mm depth error
>
> [C3] ZED 2 camera ([https://www.generationrobots.com/media/zed2-camera-datasheet.pdf](https://www.generationrobots.com/media/zed2-camera-datasheet.pdf)) — 30 mm depth error
>
> [C4] Chen et al., Accurate and Robust 3D Facial Capture Using a Single RGBD Camera, ICCV 2013.
>
> **4. Duplicated sentence in page 5.**
>
> Thanks for the detailed feedback on our writing. In the revised draft, we have modified the sentence accordingly (Sec. 3.2, page 5).

---

### Review · Reviewer_D2De · 2024-05-24

**Summary Of Contributions:**

This work presents NeuFace: a new approach for automated labeling of 3D face images with 3D face meshes. Generally, morphable parametric 3D face models such as FLAME can directly be optimized to 2D face landmarks. Instead of this, NeuFace proposes to optimize w.r.t the weights of a neural network that's trained to predict FLAME parameters from an image, rather than the parameters themselves. The paper provides analysis and quantitative evidence for the effectiveness of doing this w.r.t. neural network weights.

In addition, the paper proposes loss terms for temporal smoothness and multi-view consistency, improving the performance in multi-view and video scenarios. NeuFace is used to generate pseudo-ground truth 3D meshes for large scale 2D video datasets and is shown to improve the performance of the DECA face mesh regressor and allows for training models for long-term 3D face motion generation.

**Audience:**

Yes

**Broader Impact Concerns:**

The current ethics statement is sufficient.

**Claims And Evidence:**

Yes

**Requested Changes:**

The draft is in good shape. I would encourage the authors to include the ablations and quantitative results for motion generation in the main draft rather than in the appendix.

**Strengths And Weaknesses:**

Strengths
- The paper is well written and easy to follow. The supplementary video gives a good overview of the work, and the appendix contains a lot of helpful details such as runtime, ablation studies and additional quantitative results.
- Optimizing w.r.t. network parameters rather than is well motivated through (1) the provided intuition of taking advantage of input image information rather than relying entirely on 2D landmarks (2) the mathematical analysis (3) the visualizations of gradients to non-landmark parts of the face meshes. It's encouraging to see the gradients to update regions like the cheeks which do not traditionally have 2D landmarks but are important for high quality face meshes.
- When working with video/multi-view it makes sense to have additional loss terms for temporal/multi-view consistency. The ablations in the appendix demonstrate the effectiveness of these additional losses.
- Quantitative results demonstrate that NeuFace can generate higher quality 3D pseudo labels than off-the-shelf regressors or direct morphable model parameter optimization.

Weaknesses
- While the face motion generation results are encouraging, the value of the proposed approach and resulting datasets would be even more clear if this data can be used to obtain state of the art face mesh regression performance. The results in Table 3 show a slight improvement when DECA is trained on the new data, but this is not state of the art performance.

---

> ### Author Response · Authors · 2024-06-05
> **Response to Reviewer D2De**
>
> We thank the reviewer for acknowledging the contributions of our work and the valuable feedback that strengthens our paper. We address the comment below and in the revision (please check the updated pdf, highlighted in pink).
>
> Please let us know if our answers adequately address the reviewer’s comments. We would be happy to provide further discussions and clarifications.
>
> **1. The results in Table 3 show a slight improvement when DECA is trained on the new data, but this is not state of the art performance.**
>
> The main focus of our proposed NeuFace optimization is to build a large-scale “3D face dataset for videos,” rather than static faces.
>
> The experiment for the “static” face reconstruction in Table 3 was conducted to demonstrate one of the potential applications of our method & dataset. Given our emphasis on video face data, our model may not obtain the best accuracy for static face reconstruction in this current form (there might be potential if we deploy more accurate component modules). However, it is important to note that our method still shows  marked improvements over the existing datasets. This underscores the effectiveness of NeuFace-dataset in enhancing 3D face reconstruction. Also, we evaluated the validity of our proposed pseudo-data annotation method in Figure 4 and Table 2 in the main draft. Please consider the main points of our work.
>
> By the reviewer’s comment, we found that the relevant experiments on monocular 3D face reconstruction including Table 3 were more emphasized than other applications in the manuscript. Therefore, we have reordered Sec. 5.1 (Improving the 3D reconstruction accuracy) and Sec. 5.2 (Learning 3D human facial motion prior) in the manuscript to address this confusion. Additionally, we have added a discussion and limitations of this experiment (Sec 6 in the revision) to tone down this part.
>
> **2. Include the ablations and quantitative results for motion generation in the main draft rather than in the appendix.**
>
> Thanks for the suggestion. We have moved the loss ablation and quantitative results for motion generation in the revised main draft accordingly (Sec. 4.2, Table 3, Figs. 6-8).
>
> With the above comments, the focus of our manuscript now becomes more apparent. Thank you for the valuable comments.

---

### Author Response · Authors · 2024-06-05

We thank the reviewers for their constructive comments. We appreciate the positive feedback acknowledged by the reviewers as:

- NeuFace dataset could be helpful for 3D tasks in academia and industry (`reviewer HkHi`).
- NeuFace optimization could be a good choice, considering its effectiveness and efficiency (`reviewer HkHi`).
- Addresses an important problem with valuable motivation (`reviewer BzRa`).
- Leveraging temporal and spatial face data leads to good results (`reviewer BzRa`).
- Well demonstrated the applications of the proposed dataset (`reviewer BzRa`).
- Well-written, and draft is in good shape with helpful details in supp video & appendix (`reviewer D2De`).
- Optimizing w.r.t. network parameters (i.e., neural re-parameterization) is well motivated with detailed explanations, mathematical analysis, and gradient visualization (`reviewer D2De`).
- Using temporal/multi-view consistency loss makes sense, and ablation demonstrates its effectiveness (`reviewer D2De`).
- NeuFace can generate higher-quality 3D pseudo labels than other labeling methods (`reviewer D2De`).
- Face motion generation results are encouraging (`reviewer D2De`).

In this rebuttal, we have addressed all the comments from the reviewers, and we look forward to additional feedback or constructive discussion for clarification. Here is the summary of the rebuttal and the revision.

- We have newly added ablation experiment to support effectiveness of our EM-style NeuFace optimization (see Appendix Sec. B and Table S3).
- We have added additional statistics related to our dataset (see Table 1 in the revision and Appendix Sec. A.).
- We have added additional discussion about the experiment (see Sec. 6 in the revision).
- We have revised and re-ordered the writing following the reviews.
- All other comments from the reviewers are addressed in the individual responses.

Please let us know if our answers address the reviewer’s concerns. We would be happy to provide further discussions and clarifications.

---

### Decision · Action_Editor_QBmj · 2024-07-21

**Recommendation:** Accept as is

**Comment:**

All reviewers were positive on the paper after the revision:
- "This work presents a novel approach for automated labelling of 3D face meshes. The key observation is to optimize with respect to network parameters rather than 2D projections of 3D landmarks, and the effectiveness of this is clearly demonstrated. The revised manuscript places an emphasis on video rather than static images, clarifying the contributions of this work."
- "Thanks for the clarification. The author's response has resolved most of my questions, and this paper attempts to address a valuable task."
- "The technical contribution of the proposed NeuFace and the dataset contribution can help to improve the 3D face model estimation. The gradient analysis is also insightful."

The paper makes a good contribution to the area of 3D face processing and analysis.

**Audience:**

Yes, the dataset could be useful for researchers working on problems involving 3d face meshes or 3D face data.

**Claims And Evidence:**

Claims are appropriate, and evidence is given in experiments.